# Non-homogenous intratumor ionizing radiation doses synergize with PD1 and CXCR2 blockade

Paul Bergeron [1], Morgane Dos Santos[2], Lisa Sitterle[1], Georges Tarlet [3], Jeremy Lavigne[3], Winchygn Liu[1], Marine Gerbé de Thoré[1], Céline Clémenson[1], Lydia Meziani [1], Cathyanne Schott[1], Giulia Mazzaschi[1], Kevin Berthelot [1], Mohamed Amine Benadjaoud[4], Fabien Milliat[3,5], Eric Deutsch [1,5] & Michele Mondini [1] ✉

The efficacy and side effects of radiotherapy (RT) depend on parameters like dose and the volume of irradiated tissue. RT induces modulations of the tumor immune microenvironment (TIME) that are dependent on the dose. Low dose RT (LDRT, i.e., single doses of 0.5–2 Gy) has been shown to promote immune infiltration into the tumor. Here we hypothesize that partial tumor irradiation combining the immunostimulatory/non-lethal properties of LDRT with cell killing/shrinkage properties of high dose RT (HDRT) within the same tumor mass could enhance anti-tumor responses when combined with immunomodulators. In models of colorectal and breast cancer in immunocompetent female mice, partial irradiation (PI) with millimetric precision to deliver LDRT (2 Gy) and HDRT (16 Gy) within the same tumor induces substantial tumor control when combined with anti-PD1. Using flow cytometry, cytokine profiling and single-cell RNA sequencing, we identify a crosstalk between the TIME of the differentially irradiated tumor volumes. PI reshapes tumor-infiltrating CD8+ T cells into more cytotoxic and interferon-activated phenotypes but also increases the infiltration of pro-tumor neutrophils driven by CXCR2. The combination of the CXCR2 antagonist SB225002 with PD1 blockade and PI improves tumor control and mouse survival. Our results suggest a strategy to reduce RT toxicity and improve the therapeutic index of RT and immune checkpoint combinations.

As a major weapon against cancer, radiotherapy (RT) is approximately used for more than 50% of all cancer patients during their course of illness[1]. RT induces direct tumor cell killing through the generation of DNA damage, but RT also affects the tumor microenvironment (TME), leading to profound changes in the anti-tumor immune response. IR induces immunogenic cell death (ICD), making tumor-associated antigens (TAAs) available for priming of tumor specific cytotoxic T lymphocytes (CTLs)[2]. Moreover, IR can stimulate the interferon (IFN) cascade through the activation of the STING DNA-sensing pathway[3], promoting the maturation of DCs and their antigen presentation activity,

[1]Gustave Roussy, INSERM U1030, Université Paris-Saclay, Villejuif, France. [2]Institut de Radioprotection et de Sûreté Nucléaire (IRSN), PSE-SANTE/SERAMED/LRAcc, Fontenay-aux-Roses, France. [3]Institut de Radioprotection et de Sûreté Nucléaire (IRSN), PSE-SANTE/SERAMED/LRMed, Fontenay-aux-Roses, France. [4]Institut de Radioprotection et de Sûreté Nucléaire (IRSN), PSE-SANTE/SERAMED, Fontenay-aux-Roses, France. [5]These authors contributed equally: Fabien Milliat, Eric Deutsch. ✉e-mail: michele.mondini@gustaveroussy.fr

contributing to the amplifications of an anti-tumor adaptive immune response. By triggering several features of systemic anti-tumor immunity, IR has been demonstrated to induce anti-tumor responses in lesions outside of the irradiation field in patients[4,5]. This phenomenon, called "abscopal effect", although a rare outcome, has encouraged multiple efforts into harnessing the local and systemic effects of RT. With the emergence of immunotherapies, the interest for combining RT and immunomodulators has grown in recent years. Beneficial effects in clinical trials have been described, notably with the use of anti-PD1 and anti-CTLA4 immune checkpoint inhibitors (ICIs)[6–10].

The ability of RT to eliminate cancer cells relies on different technical parameters, such as the prescribed radiation dose, the distributed dose to the tumor volume and the volume of normal tissue inevitably irradiated[11]. However, these parameters are also known to be critical for radiation-induced side effects. Indeed, surrounding healthy tissues still undergo cell damage eventually leading to normal tissue toxicity. These side effects can drastically limit the therapeutic index of the treatment by inducing normal tissue injury in organs at risk in the radiation field, which may vary depending on both the tumor localization and the type of normal tissue exposed[12]. Over the past decades, the delivery accuracy of the irradiation (IR) dose has greatly improved[13]. The development of stereotactic body radiation therapy (SBRT) is one of the most significant advances in modern radiotherapy, and a treatment of choice in several settings[14]. By using accurate target delineation, motion management, conformal treatment planning, and daily image guidance, SBRT can deliver high doses in a few fractions and provide a steep dose fall-off outside the target[15]. However, despite substantial improvement in this area in recent years, radiation-related toxicities are still common, sometimes limiting the prescription of tumor curative dose to tumors close to organs at risk when dose/volumes of normal tissue irradiated for critical organs are above tolerance thresholds. Several efforts have been made to limit radiation side effects, precision's radiotherapy technology developments have increased ballistic accuracy and optimized the gradient between normal tissue and tumor dose de-escalation strategies have also been proposed conveying the potential risk of under effective tumor cell kill.

Beside direct tumor cell killing, experimental approaches exploring the impact of IR dose reduction have enlightened the interesting effects of low dose-radiation therapy (LDRT) on the TME. LDRT could enable the recruitment of tumor-reactive effector T cells, but also trigger the polarization of M2-like macrophages into M1-like phenotypes and increase tumor vascularization[16]. LDRT improves TME's immune infiltration, favoring responsiveness to combinatorial immunotherapy[17]. These data suggest that low IR doses could be used to promote an RT-induced anti-tumor immune activation. In agreement, immune activation was observed in bilateral tumor settings with the primary tumors irradiated with high dose-radiation therapy (HDRT) and the secondary tumors with LDRT in different settings[18–21].

In contrast to dose de-escalations, few approaches have been tested regarding the reduction of irradiation volumes[22], mainly in clinical settings[23]. Preliminary observations made in patients with large, advanced solid tumors suggested that the reduction of tumor irradiated volume was not significantly affecting the efficacy of a combination of SBRT + anti-PD1[24], followed by a phase 1 study showing the safety of partial tumor irradiation combined with pembrolizumab[25].

We hypothesized that performing image-guided partial tumor irradiations combining the immunostimulatory/non-lethal properties of LDRT with cell killing/tumor shrinkage properties of HDRT within the same tumor mass would prompt an anti-tumor effect when combined with appropriate ICI. This modality of spatially fractionated RT (SFRT) might offer the potential to reduce RT-related toxicities by reducing the volume of tissue irradiated at high doses and theoretically combine both the effects of optimal tumor cell killing and TME reshaping by taking advantage of different dose response on those two compartments (i.e., the tumor and TME).

In this work, we performed irradiations with millimetric precision to deliver different IR doses within the same tumor mass in murine colorectal and breast cancer models, using female mice. We find that the combination of PD1 blockade with LDRT and HDRT within the same tumor volume induces substantial tumor control. We decipher the influence of partial irradiation on the immune microenvironment of the differentially irradiated tumor volumes. Our results show crosstalk between the differentially-treated areas of the tumor, notably illustrated by the increased infiltration of CD8+ T cells and neutrophils, along with their peculiar phenotypic shifts observed in partially-irradiated tumors combining LDRT and HDRT. Single-cell RNA sequencing analysis reveals that partial irradiation with LDRT and HDRT deeply reshapes the tumor-infiltrating CD8T cells into more cytotoxic and interferon-activated phenotypes. However, this irradiation setting also favors protumor phenotypes among tumor-infiltrating neutrophils. In this context, we identify CXCR2 as a target to reduce protumor neutrophil infiltration following partial irradiation. We demonstrate that combining SB225002 (a potent, selective and non-peptide CXCR2 antagonist) with PD1 blockade and partial irradiation (LDRT + HDRT) further improves tumor control and mouse survival.

## Results

### The combination of high and low irradiation doses within the same tumor mass and anti-PD1 are effective against murine colorectal and breast tumors

We established experimental murine tumor models that allowed us to perform partial tumor irradiation with millimetric precision. Immunocompetent C57BL/6 mice were injected subcutaneously with MC38 tumor cells, and the total and partial tumor irradiations were performed using a Small Animal Radiation Research Platform (SARRP). As depicted in the schematic diagram in Fig. 1A, we performed irradiations of the whole tumor volume (total irradiation, TI) at high dose (16 Gy, TI16) or low dose (2 Gy, TI2), or irradiations of 50% of the tumor volume (partial irradiation, PI) at 16 Gy (PI16/0), and a combination of 50% of tumor volume irradiated at 16 Gy with the other 50% irradiated at 2 Gy (PI16/2). The techniques used to perform PI, including the combination of high and low doses (segmentations shown in Fig. 1B), showed optimal dose distributions (Fig. 1C). We biologically validated the precision and reproducibility of the PI by performing γH2AX immunohistochemistry. The tumor was oriented thanks to a two-ink staining at the time of tumor collection (Fig. 1D) and γH2AX positivity was restricted to the target irradiated volume (Fig. 1E). The ink staining also allowed us, for some experimental settings, to separate the two parts of PI tumors (Fig. 1D), which were subsequently analyzed independently. The reliability of such a procedure is demonstrated by γH2AX ELISA, which showed similar levels of Ser139 phosphorylation in tissues from PI tumors when compared to their TI counterparts (Fig. 1F).

We next evaluated the efficacy of the different IR regimens in combination with anti-PD1 or control IgG, against murine colorectal (MC38 and CT26) and triple negative breast (4T1) tumors. TI at 16 Gy resulted in a strong anti-tumor response, with complete tumor regressions in 70% of the mice bearing subcutaneous MC38 tumors and extended survival (Fig. 2A–C). In contrast, PI16/0 did not significantly slow tumor growth and did not improve mouse survival. The combination of high dose and low dose (PI16/2) induced a slight growth delay and only weakly improved the mouse survival. When mice were treated with PD1 blocking antibodies (Supplementary Fig. 1A), non-irradiated mice and mice treated by PI16/0 showed a weak response, while the combination of HDRT and LDRT (PI16/2) with anti-PD1 exerted a strong anti-tumor activity (Fig. 2A–C), with a significantly improved survival (median survival: 21 days for control mice, median survival was not reached in PI16/2+anti-PD1 group, Fig. 2C). The efficacy of PI16/2 with anti-PD1 was even higher in the

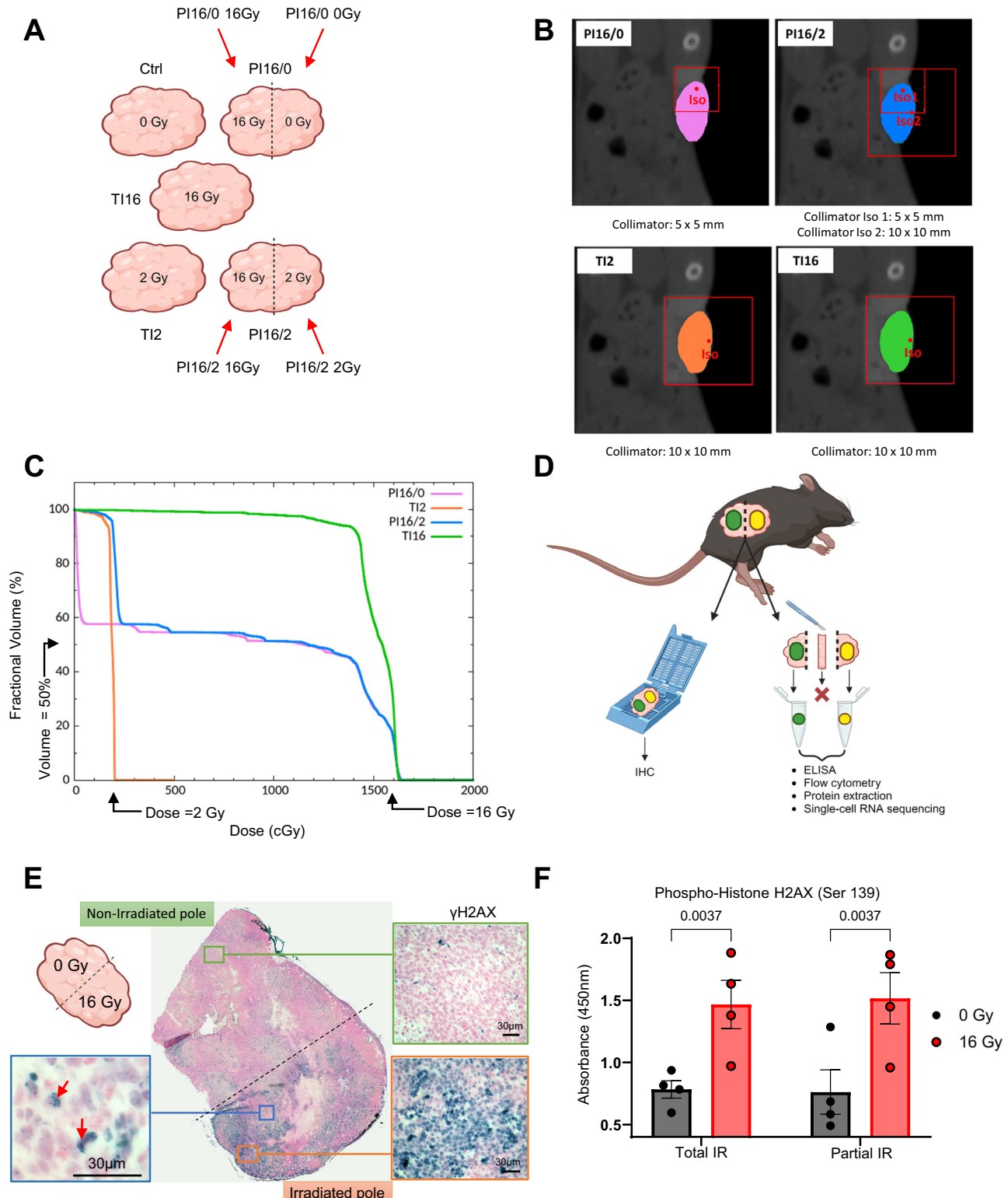

CT26 model with nearly 90% of the mice that had complete tumor clearances (Fig. 2D, E and Supplementary Fig. 1B). We further validated the efficacy of intratumor dose modulation in combination with anti-PD1 in an orthotopic setting, using the poorly immunogenic 4T1 cell line. As expected, the efficacy of all treatments involving anti-PD1 were lower than in the colorectal setting, with no efficacy of ICI alone or in combination with LDRT (Supplementary Fig. 1D, E). In contrast, PI16/2 + anti-PD1 exhibited significant anti-tumor activity and extended survival. The combination of LDRT and HDRT, but not

PI16/0, sensitized 4T1 tumors to PD1 blockade, with an increased survival in PI16/2 + anti-PD1 when compared to PI16/2 irradiation alone.

### The immune environment of tumors partially-irradiated with high and low doses presents distinctive features

With the goal to further improve the efficacy of the combination of PI with ICIs, and to investigate the mechanisms underlying the efficacy of PI16/2 combined with anti-PD1, we analyzed the modulation of the

**Fig. 1 | Methodological set-up and biological validation of non-homogenous tumor irradiation in immunocompetent mice bearing colorectal tumors.**
**A** Schematic representation of the different types of RT treatments, involving control tumors (Ctrl, 0 Gy), total tumor volume 2 Gy RT (TI2), total tumor volume 16 Gy RT (TI16), partial tumor volume 16/0 Gy RT (PI16/0), and partial tumor volume 16/2 Gy RT (PI16/2). The different portions of partially-irradiated tumors are also identified. Portions irradiated at 16 Gy and 0 Gy (unirradiated) from PI16/0 tumors (top right of the panel) are respectively called PI16/0 16 Gy and PI16/0 0 Gy. Reciprocally, portions irradiated at 16 Gy and 2 Gy from PI16/2 tumors (bottom right of the panel) are respectively called PI16/2 16 Gy and PI16/2 2 Gy. Black-dotted lines represent the limit between the two halves of partially-irradiated tumors. Created in BioRender. Mondini (2024) BioRender.com/j74g766. **B** Representative tumor segmentations generated with the help of CT imaging, representing all the irradiation regimens used in this article (TI2, TI16, PI16/0, PI16/2). The collimator (red squares in the figure) used for each isocenter (Iso, red points in the figure) is indicated below the associated panel. **C** Dose-volume histogram (DVH) showing dose distribution within the tumor volume, depending on the RT modality (dose, TI or PI) of the segmentations presented in (**B**). **D** Schematic representation of tumor staining and sampling method depending on the read-out. Both extremities of the tumors were identified and separated according to the procedure described in "Methods". The intermediate portion (unstained) separating the extremities (green and yellow) was discarded. Created in BioRender. Mondini (2024) BioRender.com/n07j238. **E** Representative image of a γH2AX (dark blue) staining performed following partial tumor volume irradiation (PI16/0, 16 Gy and 0 Gy) on subcutaneous MC38 mouse tumors ($n = 4$ mice, two independent experiments). The red arrows show γH2AX foci. **F** H2AX phosphorylation on serine 139 was quantified by ELISA in Ctrl tumors ("Total IR", 0 Gy), in TI16 tumors ("Total IR", 16 Gy), and in both halves of PI16/0 tumors ("Partial IR", 16 Gy and 0 Gy), with $n = 4$ mice in all groups. Data are shown as the mean ± SEM. Numbers on these graphs represent $p$ values and were determined by two-way ANOVA. Source data are provided as a Source Data file.

MC38 tumor immune environment induced by the different irradiation treatments. Two days after irradiation, when tumor growth curves of the different treatment groups started to diverge (Fig. 2B, C), we collected the tumors and separated the differently irradiated tumor volumes (Fig. 3A) following the procedure described in Fig. 1D. Flow cytometry analyses showed slightly reduced CD8+ T cells levels in TI tumors both at 2 and 16 Gy compared to control samples, which was not observed in PI tumors (Fig. 3B). Of note, in the tissue from PI16/2 that received 16 Gy, CD8+ T cell density was significantly higher than in TI16 tumors. We observed a significant increase in NK cells density in the 16 Gy volume from PI16/2 tumors, both compared to control and to TI16 specimens (Fig. 3B). Regarding the myeloid compartment, neutrophils were increased in all volumes that received 16 Gy irradiations, both in TI and PI samples, reaching significance in the PI16/2 sample. In the 2 Gy volume of PI16/2 tumors, neutrophil levels trended towards higher levels than in TI2, suggesting an impact of the proximity of the 16 Gy tumor volume. The neutrophils from both volumes of PI16/2 tumors expressed high levels of immunosuppressive surface molecules such as PD-L1 and CD206 (Supplementary Fig. 2B). An increase of regulatory T cells (Tregs) and monocytes was also observed in the 16 Gy volume of PI16/2, when compared to TI16 samples (Supplementary Fig. 1B, C).

We next analyzed the cytokine/chemokine profile of the different samples. sPLS-DA analysis using 35 cyto/chemokines as variables showed that most of the samples from 16/2 irradiated tumors were projected in the same sPLS-DA regions than the TI16 and TI2 samples respectively, but in specific sub-regions (right for PI16/2–16 Gy and top for PI16/2–2 Gy) (Fig. 3C), suggesting that PI impacted the cytokine environment differently from TI, either at low or high IR doses. CCL5 was significantly increased in all irradiated tumor tissues, with the highest concentration observed in PI16/2 samples both in the 16 Gy and 2 Gy volumes (Fig. 3D). The other CCR5 ligand (CCL4) was significantly increased in samples IR at 16 Gy either from PI or TI tumors. We also observed a significant increase in the CXCR2 ligand CXCL1 in the 16 Gy volume from PI16/2 tumors (Fig. 3D). The highest levels of another CXCR2 ligand, CXCL2, were also observed in the 16 Gy volume of PI tumors.

In order to deeply characterize the phenotypic landscape of the immune cells, we performed single cell RNA sequencing on CD45+ cells from the different samples. Following unsupervised clustering, we identified the main lymphoid and myeloid immune populations according to their gene expression profiles (Fig. 4A, B). The proportion of neutrophils strongly increased in samples irradiated at 16 Gy, and in particular in PI tumors (Fig. 4C). Of note, in the volume irradiated at 2 Gy of PI16/2 tumors, the increase in the proportion of neutrophils was markedly higher than in TI2 tumors. These data, together with the flow cytometry and cytokine profiling analyses, demonstrates that PI at 16/2 Gy induces deep changes in the tumor immune environment.

## CD8+ T cells are phenotypically re-shaped following partial irradiation, and are critical in mediating its anti-tumor activity

CD8+ T cells are known effectors of the IR-induced anti-tumor immune response, as well as one of the main targets of ICIs as anti-PD1. Moreover, their levels were found to be particularly increased in PI16/2 tumors. We thus performed a deep characterization of their phenotype and its modulation upon the different IR schemes. We performed an unsupervised sub-clustering of the CD8+ cluster identified by scRNA-seq, detecting six main CD8+ subpopulations presenting distinctive expression profiles (Fig. 5A). By analyzing their transcriptomic signature, we identified each cluster as corresponding to a specific CD8+ T cell phenotype (Fig. 5C and Supplementary Fig. 3A), including proliferating, naive, short live effector, IFN-activated, pre-memory effector and central memory CD8+ cells (Supplementary Fig. 3B). Pseudotime analysis showed that, in our tumor model, CD8+ T cells progressed from their naïve state to an interferon (IFN)-activated phenotype, with their evolution then directed either towards memory functions or to a short-lived fate with a phenotype presenting signs of exhaustion (Fig. 5B), including the expression of immune checkpoints as PD1 and LAG3 (Supplementary Fig. 3A). When we analyzed the distribution of CD8+ T cells from the different samples in the six sub-clusters, we observed a strong increase in the proportion of IFN-activated cells in samples from 16/2 tumors in both the 2 Gy and 16 Gy volumes, with a concomitant reduction of the short-lived cluster, when compared to control tumors (Fig. 5C). This phenotype shift was not observed in the other irradiated samples, except in TI2 ones but to a lower extent. In agreement with the observed shift towards an activated state, CD8+ T cells from PI16/2 tumors presented the highest type IFN activation score, together with TI2 samples (Fig. 5D), and a similar picture was observed concerning their cytotoxicity score (Fig. 5E). These data were supported by flow cytometry results suggesting that combining PD1 blockade to PI16/2 irradiation could increase the infiltration of CD8+ T cells expressing IFNγ and Granzyme B (Supplementary Fig. 3C), at least in the 2 Gy portion of the tumor. Of note, CD8+ T cells from the 16 Gy and 2 Gy irradiated portions of PI16/2 tumors showed very similar expression profiles in the scRNA-seq data, with few differentially modulated genes, in contrast to the large number of differentially modulated genes identified when comparing TI16 vs. TI2 samples (compare Supplementary Fig. 3D to Supplementary Fig. 3E).

Given the specific phenotype observed in PI16/2 tumors compared to TI tumors (Fig. 5C–E) and that CD8+ T cells are targets of PD1 blocking antibodies, we sought to determine a functional role of CD8+ T cells in the anti-tumor response to 16/2 partial irradiations, alone and in combination with anti-PD1. When we depleted CD8+ T cells, the response to partial irradiations was completely impaired, both in terms of tumor sizes (Fig. 6A, C) and mouse survival (Fig. 6B), including in groups treated with anti-PD1 antibodies. Conversely, CD8+ T cells were partly dispensable in TI16 irradiation settings, as we observed a delayed tumor growth and a significantly increased survival in the TI16 + anti-CD8 group, when compared to control mice (Fig. 6A–C).

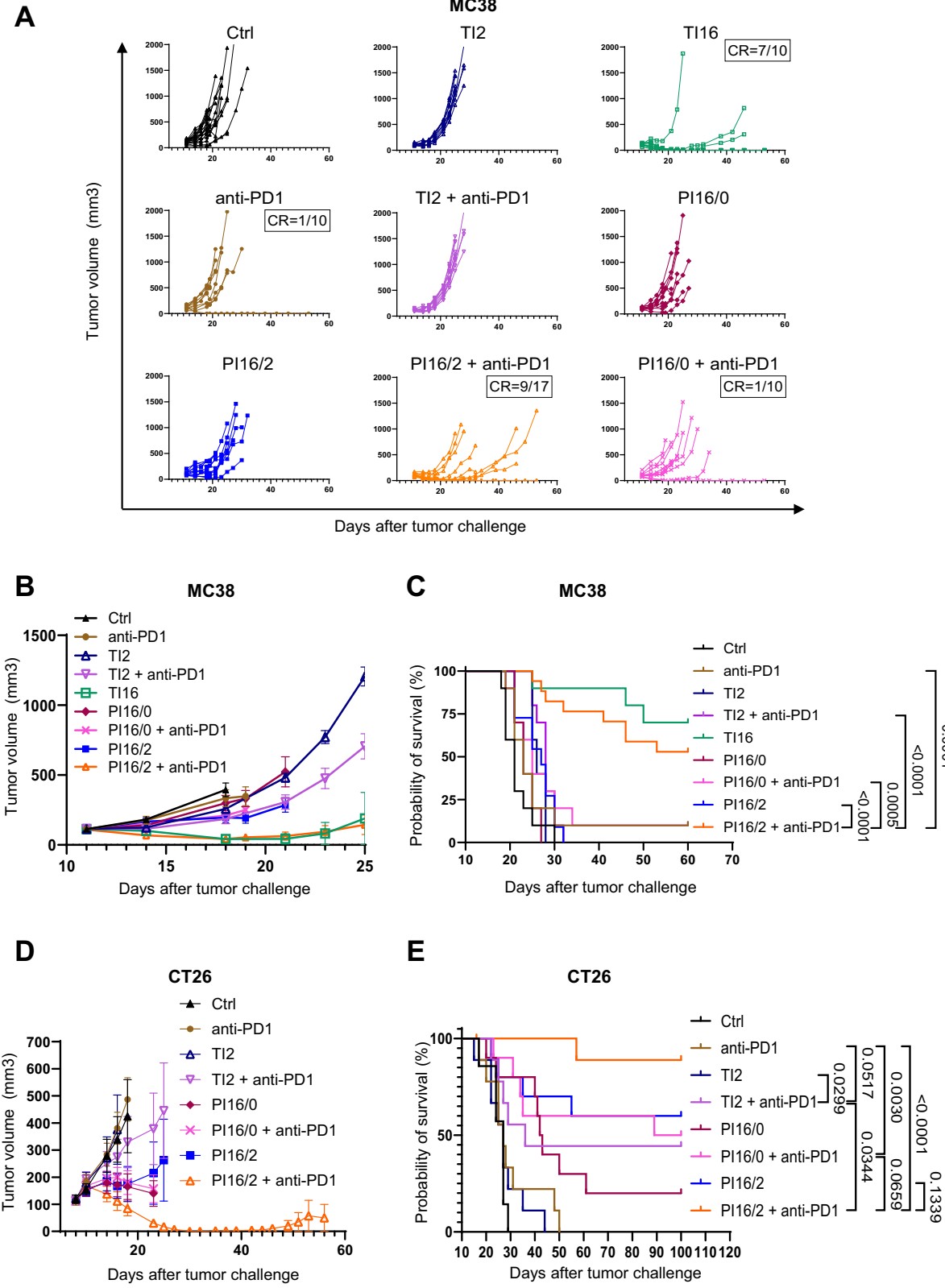

**Fig. 2 | Partial tumor irradiation combining LDRT and HDRT with anti-PD1 improves tumor control and survival in immunocompetent mice bearing colorectal tumors. A** Individual curves of subcutaneous MC38 tumor growth in C57BL/6 mice (Ctrl $n = 18$, TI2 $n = 10$, TI16 $n = 10$, anti-PD1 $n = 10$, TI2 + anti-PD1 $n = 10$, PI16/0 $n = 10$, PI16/0 + anti-PD1 $n = 10$, PI16/2 $n = 11$, PI16/2 + anti-PD1 $n = 17$; combined from three independent experiments). CR complete response. **B** Mean tumor volume growth of subcutaneous MC38 cells in C57/Bl6 mice from (**A**), starting on the day of irradiation. Curves were stopped when the first mouse euthanasia occurred. **C** Kaplan−Meier survival curves of the efficacy experiment

shown in (**A**, **B**). **D** Mean tumor volume growth of subcutaneous CT26 cells in Balb/ C mice, starting on the day of irradiation (Ctrl $n = 7$, TI2 $n = 9$, anti-PD1 $n = 9$, TI2 + anti-PD1 $n = 9$, PI16/0 $n = 10$, PI16/0 + anti-PD1 $n = 10$, PI16/2 $n = 10$, PI16/2 + anti-PD1 $n = 9$; combined from two independent experiments). Curves were stopped when the first mouse euthanasia occurred. **E** Kaplan−Meier survival curves of the efficacy experiment shown in (**D**). Data were represented as the mean ± SEM (**B**, **D**), and $n$ represents the number of mice/groups. Numbers on survival graphs represent $p$ values and were determined by Log-rank Mantel−Cox analysis (**C**, **E**). Source data are provided as a Source Data file.

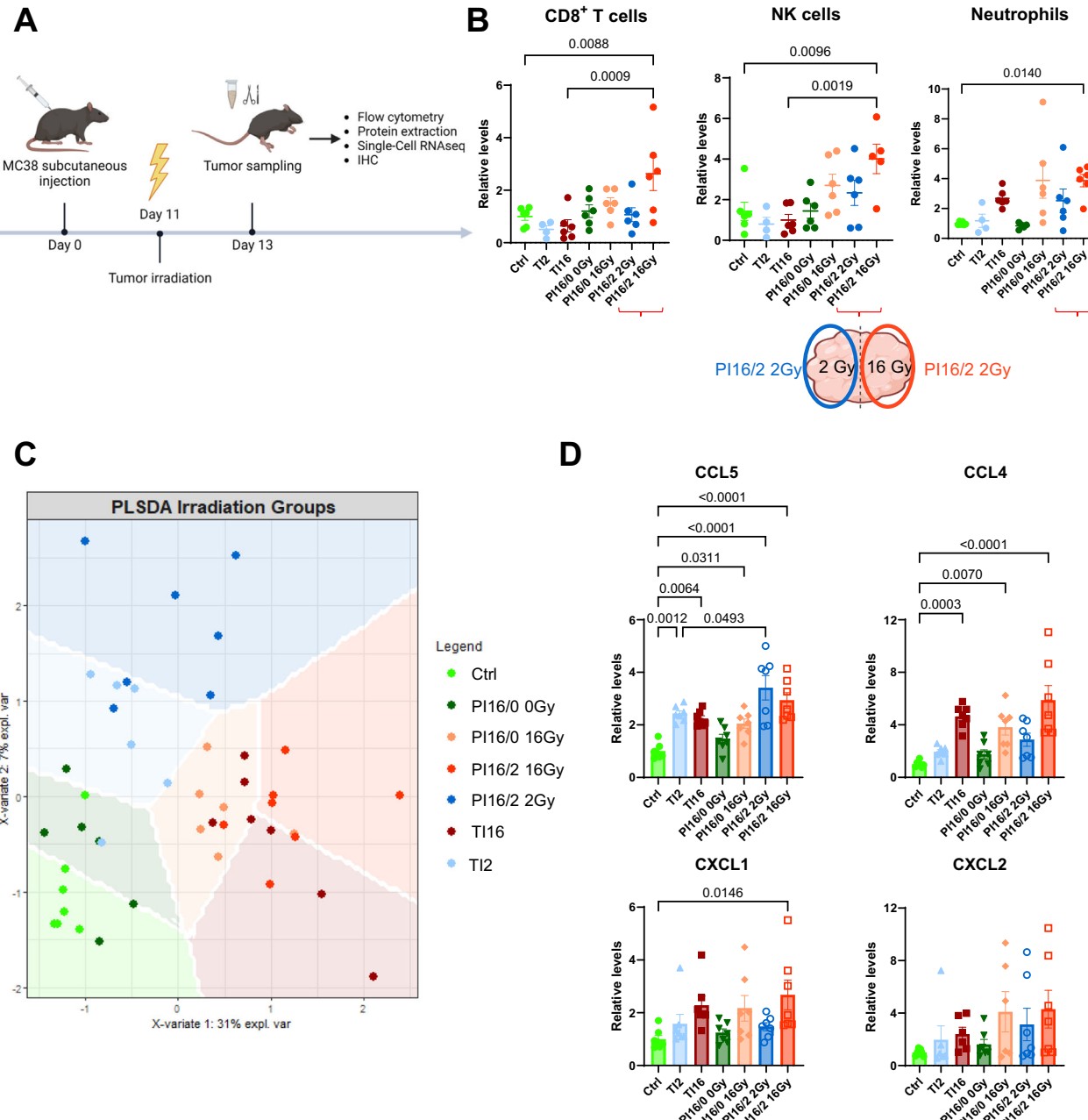

**Fig. 3 | Partial irradiation combining LDRT and HDRT within the same tumor induces specific changes in tumor immune populations and cytokines.**
**A** Setting for the experiments performed on C57BL/6 mice and shown in this figure. Created in BioRender. Mondini (2024) BioRender.com/k67d783. **B** Relative levels of number of CD8 T cells (left), NK cells (center), neutrophils (right) per mg of tumor. Relative levels were calculated by comparing each individual value with the mean cell counts in Ctrl for each population. Differentially-treated parts of partially-irradiated tumors were separated and considered as different groups. Groups representing the two differentially-treated parts of PI16/2 tumors are indicated with red brackets, and illustrated using a schematic PI16/2 tumor (PI16/2 16 Gy in red, PI16/2 2 Gy in blue). Data were obtained from two independent experiments (Ctrl $n = 6$, TI2 $n = 4$, TI16 $n = 6$, PI16/0 0 Gy $n = 6$, PI16/0 16 Gy $n = 6$, PI16/2 2 Gy $n = 6$, PI16/2 16 Gy $n = 6$). Outliers were identified using the ROUT test and excluded from the analysis. Schematic image created in BioRender. Mondini (2024) BioRender.com/j74g766. **C** Subcutaneous MC38 tumors from C57BL/6 mice were sampled as described in (**A**) and a cytokine profiling was performed with the same

treatment groups as in (**B**). Partial Least-Squares Discriminant Analysis (PLS-DA) was applied to the cytokine/chemokine concentrations obtained for each individual of each treatment group. Data were obtained from two independent experiments ($n = 7$ in all treatment groups). **D** Relative concentration levels in pg/μL of selected cytokines/chemokines from the cytokine profiling presented in (**C**). Relative concentration levels were calculated by comparing each individual value with the mean concentrations in Ctrl for each cytokine/chemokine ($n = 7$ in all treatment groups, except $n = 6$ for the TI16, TI2 and PI16/0 16 Gy groups in the CXCL2 panel, as one measure was considered out of the dynamic range of the assay). Data were represented as the mean ± SEM (**B**, **D**), and $n$ represents the number of mice/groups. Parametric statistics were only applied to normally distributed data. Numbers on these graphs represent $p$ values and were determined by ordinary one-way ANOVA with the Šidák's multiple comparison test (**B**: CD8⁺ T cells and NK cells and **D**: CCL4, CCL5, CXCL1), and Kruskal–Wallis test with Dunn's multiple comparison test (**B**: neutrophils and **D**: CXCL2). Source data are provided as a Source Data file.

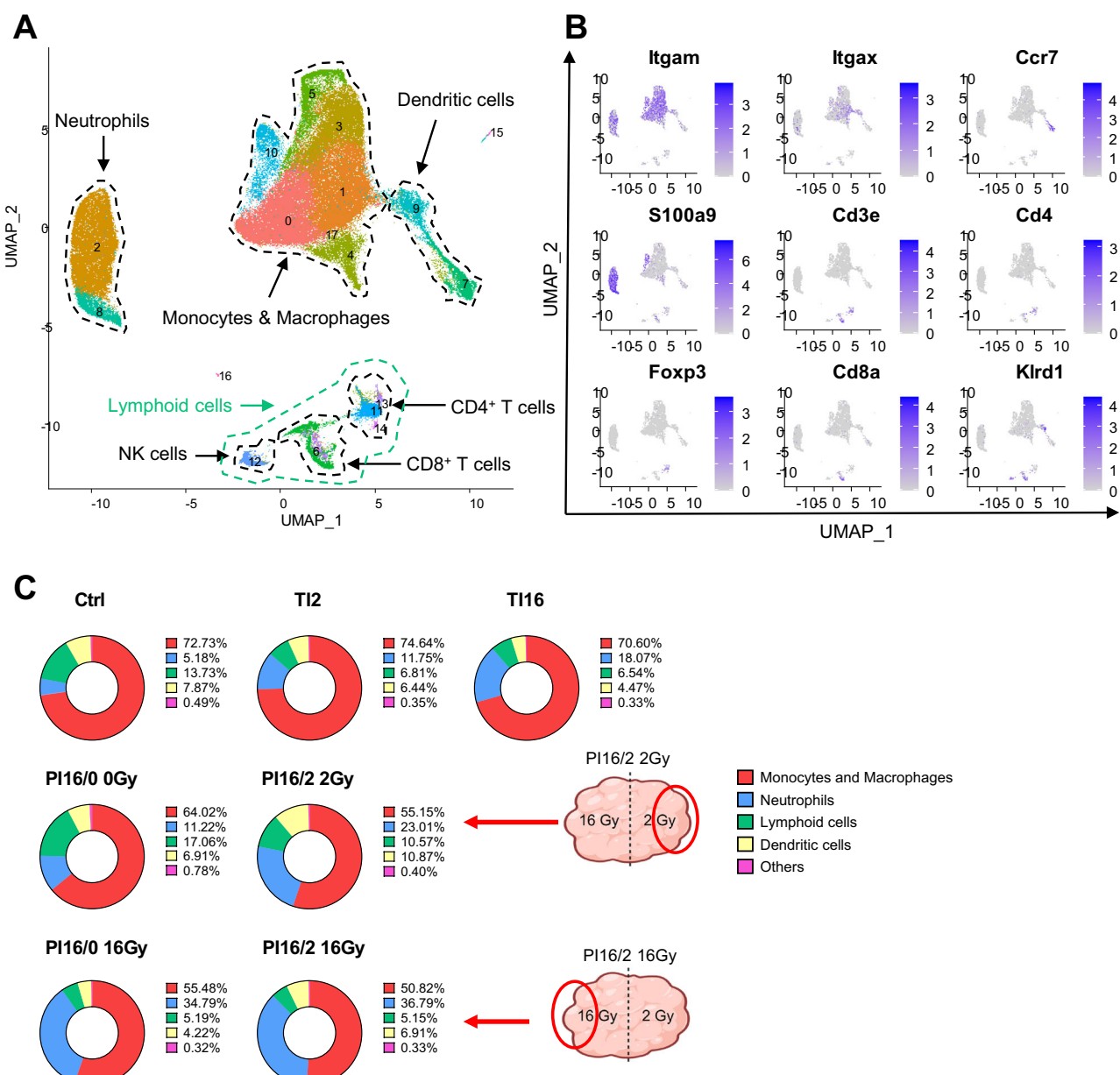

**Fig. 4 | Single cell RNA-seq analysis reveals that partial irradiation combining LDRT and HDRT within the same tumor induces deep changes in the tumor immune microenvironment. A** Subcutaneous MC38 tumors from C57BL/6 mice were treated and sampled as described in Fig. 2A, and CD45+ cells were sorted for scRNA-seq. Global uniform manifold approximation projection (UMAP) embedding of merged scRNA-seq profiles from all of the treatment groups (Ctrl $n = 4$, TI2 $n = 2$, TI16 $n = 2$, PI16/0 0 Gy $n = 2$, PI16/0 16 Gy $n = 2$, PI16/2 2 Gy $n = 2$, PI16/2 16 Gy $n = 2$; combined from two independent experiments, $n$ representing the number of mice/group) was generated ($n = 105,462$ cells in total). Each Seurat object was associated with a treatment group and was downsampled to contain $n = 15,066$ cells, in order to have equally-contributing treatment groups in the global UMAP (resolution $r = 0.5$). **B** UMAP visualizations of *ITGAM, ITGAX, CCR7, S100A9, CD3E, CD4, FOXP3, CD8A,* and *KLRD1* mRNA expression at the single-cell level in the total population, generated with the FeaturePlot function. These genes were used to identify the immune cell types shown in (**A**). **C** Pie charts representing the proportions of the immune cell types represented and identified in the global UMAP shown in (**A**) and (**B**). Each pie chart represents one treatment group ($n = 15,066$ cells). Both portions of the PI16/2 tumors are indicated with schematic tumors and red arrows. Schematic images created in BioRender. Mondini (2024) BioRender.com/j74g766. Source data are provided as a Source Data file.

Altogether, these data indicate that CD8+ T cells are pivotal mediators of the efficacy of the combination of partial irradiation and ICIs.

### Targeting immature neutrophil infiltration in partially-irradiated tumors through CXCR2 blockade improves the response to anti-PD1

As for CD8+ T cells, we observed that the number and the proportion of neutrophils were particularly elevated in PI16/2 tumors (Figs. 3B and 4C). Neutrophils can acquire different characteristics in the tumor environment, contributing either to proinflammatory or to immuno-suppressive activities. An unsupervised analysis of the neutrophil cluster identified by scRNA-seq revealed that these cells could be distinguished as three distinct groups, corresponding to different neutrophil phenotypes (Fig. 7A and Supplementary Fig. 4A, B). Pseudotime analysis confirmed the progression of neutrophils from an immature state to intermediate and then mature phenotype (Fig. 7B). The distribution of these three neutrophil clusters was uneven in the different irradiation groups, with a decrease of the proportion of mature

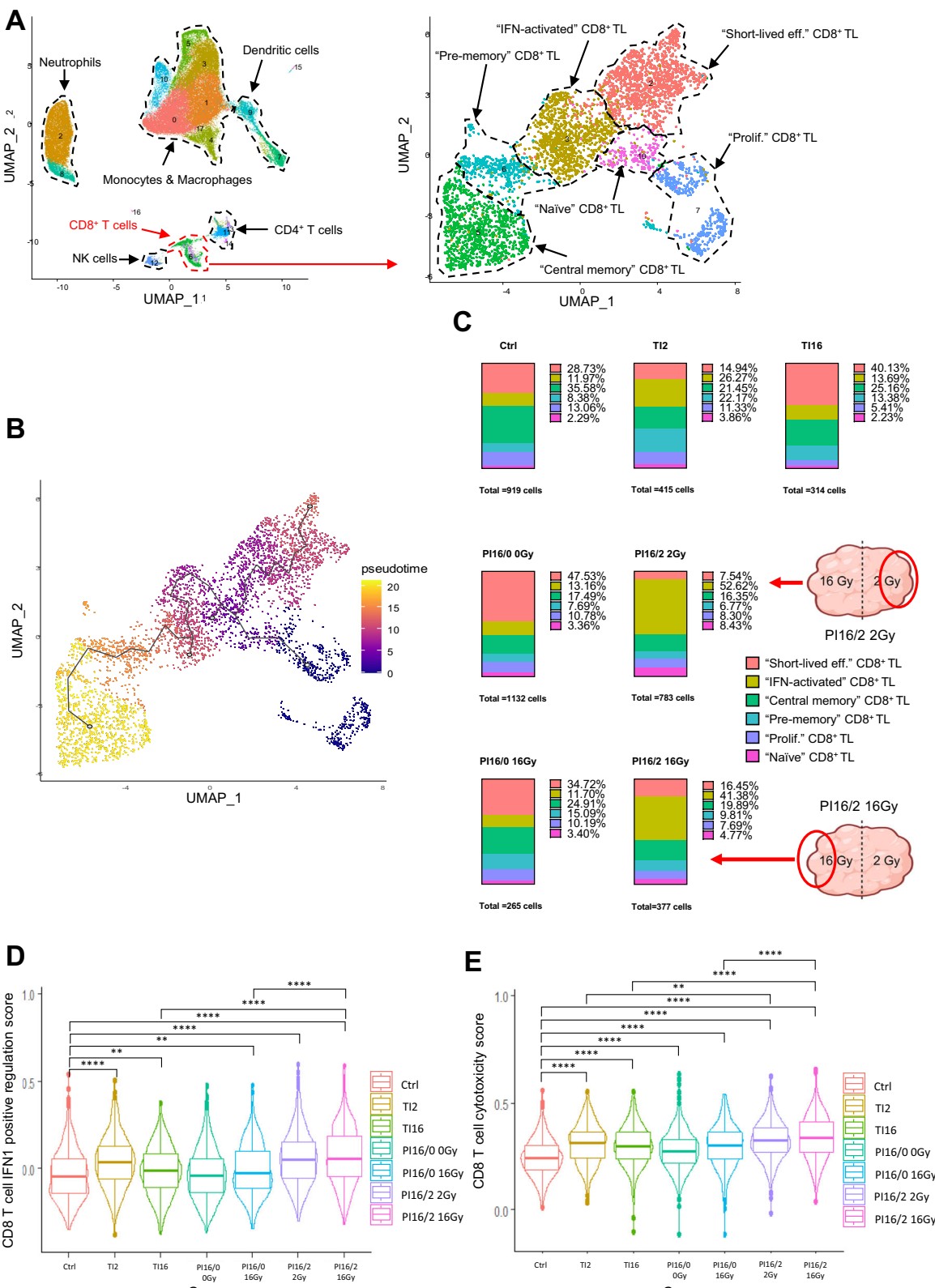

neutrophils in the PI16/2 tumors, at a particularly high extent in the 16 Gy irradiated samples from these tumors (Fig. 7C). A supervised analysis and related UMAP projections from the different groups showed that neutrophils from the PI16/2 samples were characterized by a specific transcriptional program compared with TI counterparts, and notably when comparing TI16 to PI16/2 16 Gy samples (Supplementary Fig. 4C), suggesting that PI16/2 neutrophils acquired a specific

phenotype in these samples. The neutrophil degranulation score, recapitulating their ability to release cytotoxic molecules that trigger the apoptotic elimination of cancer cells, was significantly reduced in PI16/2 tumors, with the lowest score observed in the tumor volume receiving 16 Gy (Fig. 7D). Moreover, a score that indicated the T cell suppression activity of neutrophils was significantly increased in the 16 Gy volume of the PI16/2 tumors both compared with the control

**Fig. 5 | Partial irradiation combining LDRT and HDRT reshapes the phenotype of CD8⁺ T cells within the tumor microenvironment. A** CD8⁺ T cells from the global UMAP (left, also shown in Fig. 3A) were identified (circled in dotted red, red arrow), and sub-clustering was performed to generate a new UMAP (n = 4205 cells). Differentially-expressed genes (DEGs) from each cluster were analyzed to identify CD8⁺ T cell subpopulations. Some of the DEGs that were used in this identification are available in Supplementary Fig. 3A. **B** Pseudotime analysis of CD8⁺ T cells using the Monocle package clusters isolated in (**A**). The black curve indicates the trajectory from a starting point corresponding to the proliferating CD8 T cell cluster. **C** 100% stacked columns representing proportions of CD8⁺ T cell subpopulations identified in (**A**). Number of cells for each treatment group is indicated below each associated chart. Both portions of PI16/2 tumors are indicated with schematic tumors and red arrows. Schematic images created in BioRender. Mondini (2024) BioRender.com/j74g766. Expression score levels of Type 1 IFN positive regulation (**D**) and cytotoxicity (**E**) associated gene signatures

in CD8⁺ T cell populations from the scRNA-seq experiment, depending on treatment group. Each point represents the expression level of these features (**D**; Type 1 IFN positive regulation, **E**; Cytotoxicity) at a single-cell level in the CD8⁺ T cell population. The horizontal line drawn inside of the box represents the median. The lower and upper hinges represent the first and third quartiles, respectively. The upper whisker extends from the upper hinge to the largest value within 1.5 times the interquartile range (IQR) from the hinge. Similarly, the lower whisker extends from the lower hinge to the smallest value within 1.5 times the IQR from the hinge. Data beyond the end of the whiskers are outlying points and are plotted individually. Expression scores were generated using the AddModule-Score function in Seurat. *p* values were determined by Mann−Whitney *U* test (**p < 0.01; ****p < 0.0001) and their exact values are provided as a Source Data file. Source data have been deposited in the Gene Expression Omnibus (GEO) under accession code GSE262699.

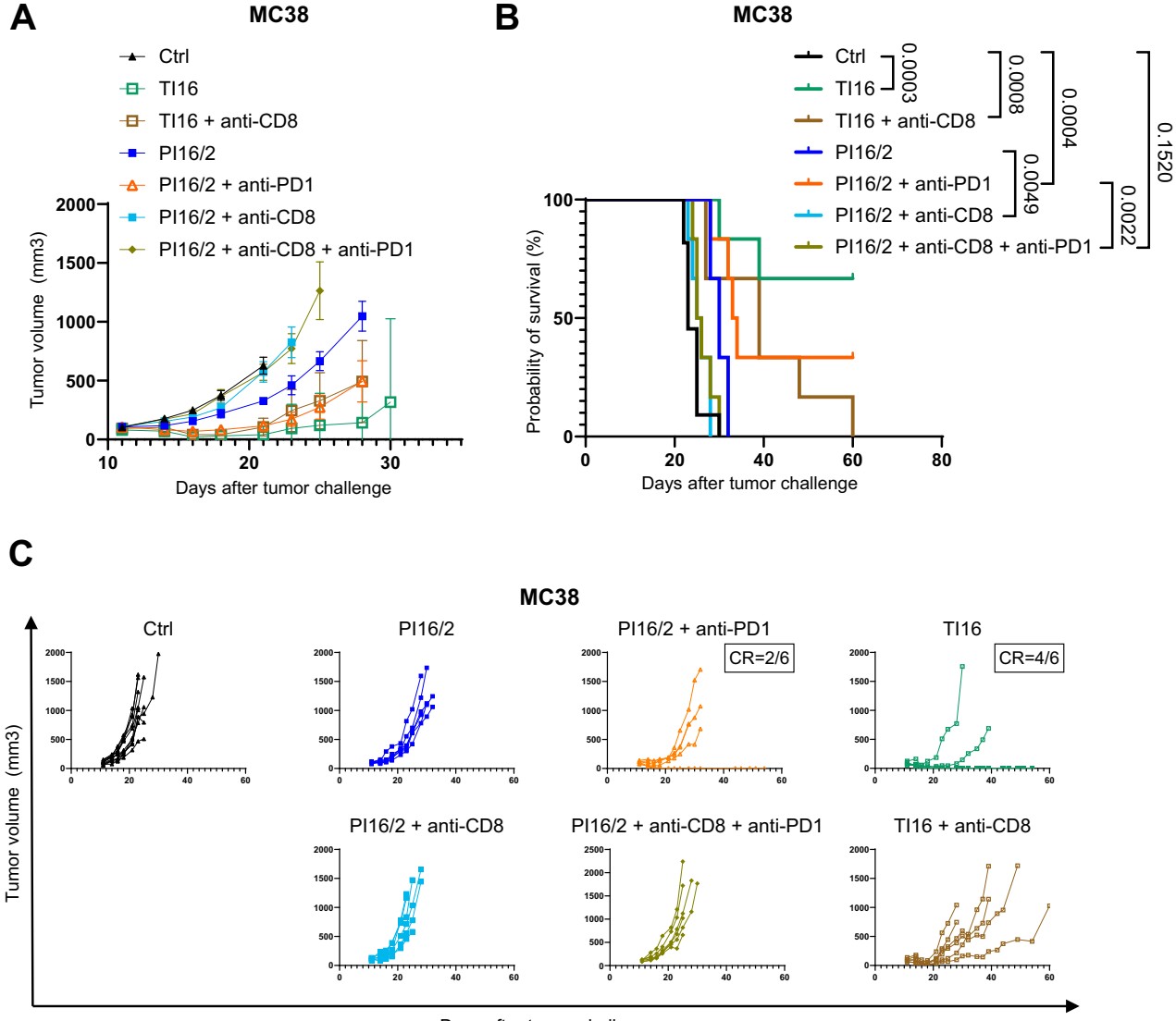

**Fig. 6 | CD8⁺ T cells play a pivotal role in the anti-tumor efficacy of partial irradiation with HDRT and LDRT combined with anti-PD1. A** Mean tumor volume growth of subcutaneous MC38 cells in C57/Bl6 mice, starting on the day of irradiation, with Ctrl n = 11 and n = 6 mice in all other groups. Curves were stopped as the first mouse euthanasia occurs in the group. Data were represented as the

mean ± SEM. **B** Kaplan−Meier survival curves of the efficacy experiment shown in (**A**). Numbers on the survival graph represent *p* values and were determined by Log-rank Mantel−Cox analysis. **C** Individual curves of subcutaneous MC38 tumor growth in C57BL/6 mice from the experiment shown in (**A, B**). Source data are provided as a Source Data file.

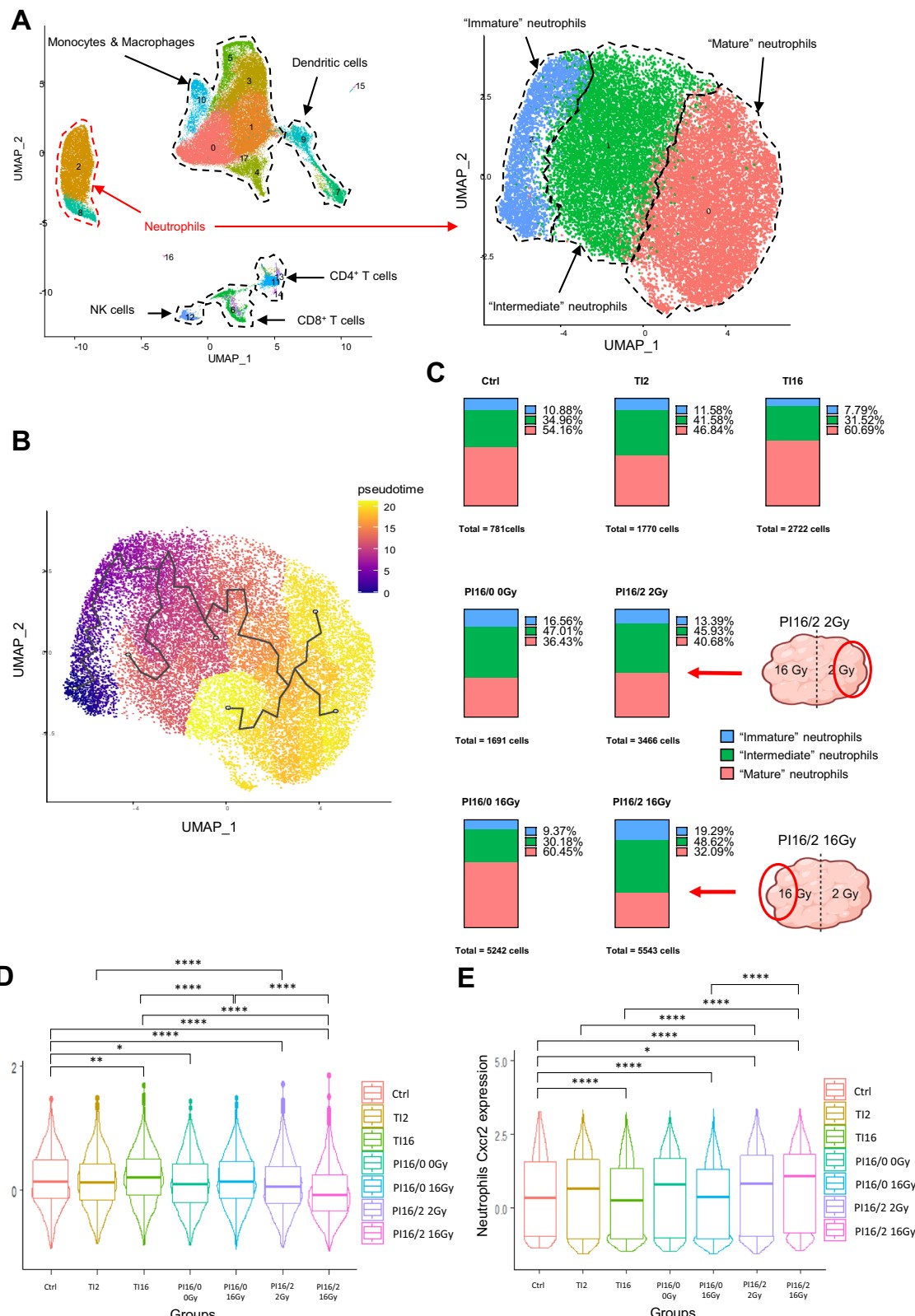

group but also with the TI16 group (Supplementary Fig. 4D). These data, together with the increased levels of immunosuppressive membrane markers observed in neutrophil from PI16/2 samples (PD-L1 and CD206, Supplementary Fig. 2B), suggested that neutrophils could contribute to immunosuppression after partial irradiations. Accordingly, their negative role in the tumor response to partial irradiation with high and low doses of RT was confirmed using Ly6G antibodies,

which depleted neutrophils (Supplementary Fig. 5A, B) and improved tumor control and mouse survival when combined to PI16/2 and anti-PD1 (Supplementary Fig. 5C–E). Neutrophil depletion was validated using Gr-1 antibodies (Supplementary Fig. 5A, B) in order to avoid the epitope masking of Ly6G[26]. Nevertheless, Ly6G expression being restricted to mice, this approach had no translational potential, and thus we sought a strategy to target the neutrophil population with a

**Fig. 7 | Partial irradiation combining LDRT and HDRT triggers the infiltration of immature neutrophils in the tumor microenvironment. A** Neutrophil clusters from the global UMAP (left, also shown in Fig. 3A) were identified (circled in dotted red, red arrow), and sub-clustering was performed to generate a new UMAP generating a new UMAP (n = 21,215 cells). Differentially-expressed genes (DEGs) from each cluster were analyzed to identify neutrophil subpopulations. Some of the DEGs that were used in this identification are available in Supplementary Fig. 4A. **B** Pseudotime analysis of neutrophils using the Monocle package. The black curve indicates the trajectory from a starting point corresponding to immature neutrophils. **C** 100% stacked columns representing proportions of neutrophil subpopulations identified in (**A**). Number of cells for each treatment group is indicated below each associated chart. Both portions of PI16/2 tumors are indicated with schematic tumors and red arrows. Schematic images created in BioRender. Mondini (2024) BioRender.com/j74g766. Expression score levels of the degranulation-associated gene signature (**D**) and the *CXCR2* gene (**E**) in neutrophil populations depending on treatment group. Each point represents the expression level of these features (**D**; Degranulation, **E**; *CXCR2*) at a single-cell level in the neutrophil population. The horizontal line drawn inside of the box represents the median. The lower and upper hinges represent the first and third quartiles, respectively. The upper whisker extends from the upper hinge to the largest value within 1.5 times the IQR from the hinge. Similarly, the lower whisker extends from the lower hinge to the smallest value within 1.5 times the IQR from the hinge. Data beyond the end of the whiskers are outlying points and are plotted individually. Expression scores were generated using the AddModuleScore function in Seurat. *p* values were determined by Mann–Whitney *U* test (*$p < 0.05$; **$p < 0.01$; ****$p < 0.0001$) and their exact values are provided as a Source Data file. Source data have been deposited in the Gene Expression Omnibus (GEO) under accession code GSE262699.

clinically druggable target. CXCR2 is a pivotal chemokine receptor for neutrophil trafficking and its expression was found predominantly in their immature cluster (Supplementary Fig. 4E). In agreement with an increased proportion of immature neutrophils in PI16/2 tumors (Fig. 7C), the highest CXCR2 expression levels were found in tumors partially irradiated with 16 Gy and 2 Gy (Fig. 7E). Flow cytometry analyses showed that the neutrophils levels in PI16/2 tumors were also increased after PD1 treatment, including the subset expressing CXCR2 (Supplementary Fig. 5F). These observations, together with the increase levels observed for the CXCR2 ligands CXCL1 (and a trend for CXCL2) in PI16/2 tumors (Fig. 3D), led us to hypothesize that, in partially irradiated tumors, CXCR2 drove neutrophil infiltration, and particularly the ones presenting an immature phenotype. To confirm this hypothesis, we treated mice with the selective CXCR2 antagonist SB225002 and we showed that this pharmacological intervention prevented the neutrophil increase observed in the 16 Gy volumes from PI16/2 tumors (Fig. 8A). CXCR2 blockade did not impact the infiltration of other immune cells, such as CD8+ T and NK cells that were even increased in PI16/2 16 Gy samples (Fig. 8A). When we combined PI16/2 with anti-PD1 and CXCR2 blockade with SB225002, tumor control (Fig. 8B, D) and survival (Fig. 8C) were further improved compared with control group (median survivals: control 23 days, PI16/2 + anti-PD1 42 days, PI16/2 + anti-PD1 + SB225002: not reached as more than 50% complete responses), suggesting that the CXCR2-recruited cells exerted immunosuppressive activities. This observation is supported by data obtained in the 4T1 orthotopic model, with the group treated with SB225002 trending towards a reduced tumor growth and a better survival, even if no complete response was observed in this poorly immunogenic model (Supplementary Fig. 6A–C).

## Discussion

The approach that we present here, combining irradiations with millimetric precision to the identification and collection of differently irradiated tumor volumes, allowed us to gain insights concerning the modulation of the TME of tumors that received non-homogenous IR doses. With this approach we found that tumors that received a high IR dose on half of their volumes combined with LDRT on the other half presented a TME that was different from the tumor totally irradiated at the same doses, in terms of cytokines, proportion of immune populations, as well as their phenotype. These specific TME modifications were associated with a synergistic anti-tumor activity with anti-PD1. We demonstrated that the combination of partial IR (LDRT + HDRT) and PD1 blockade was able to promote complete responses in murine colorectal tumors. These observations obtained with LDRT and HDRT performed within the same tumor mass may reflect the immune activation observed in abscopal settings where the primary tumors were irradiated with HDRT and the secondary tumors with LDRT[18–20], showing sign of systemic anti-tumor immune response. We selected 16 Gy as an irradiation dose that was sufficient, in a single administration, to induce complete responses in the MC38 subcutaneous

tumors. The data reported in Fig. 2A–C confirm that 70% of the mice had complete responses after a TI16 irradiation. Moreover, the non-lethal but immunostimulatory properties of LDRT in the range of 0.5–2 Gy[16,17] make the combination with HDRT an attractive strategy offering the potential to differently impact tumor cells and the TME, taking advantage of differences in dose-response between those compartments. We validated a better efficacy of PI16/2 + anti-PD1 than TI2 + anti-PD1 in all three different models, highlighting the crucial role of the ablative radiation dose in the PI setting. Of note, the results achieved in the CT26 tumor model were obtained with large tumors (tumor volume on the day of irradiation of ~140–150 mm³), suggesting that also bulky tumors could benefit from such irradiation regimens in combination with IO, in particular for "hot" tumor lesions. Further studies will be needed to define the optimal range of HDRT and LDRT to achieve a better immune activation and improve the response when combined with immunotherapy. Since LDRT at doses of 0.5 Gy or lower are used in certain non-oncologic settings to treat inflammatory and neurodegenerative conditions due to observed anti-inflammatory effects[27,28], a fine tuning of the doses of LDRT will be required to optimize the combination with immunotherapy agents. The detailed characterization of the TME of PI tumors allowed us to further improve the efficacy of PI16/2 with anti-PD1. In these tumors we observed, especially in the portions receiving HDRT from PI tumors that also received LDRT, a stronger infiltration of effector immune populations such as NK and CD8+ T cells, when compared to other conditions, and in particular to TI tumors. We also observed a trend of increase in CD8+ T (-1.4 fold) cells in the non-irradiated mass from 16/0 tumors, in line with observations made using conventional cabinet irradiations[22]. Of note, these flow cytometry data were shown in absolute cell counts per mg of tumor, as proportions do not always succeed in accounting for variations in less abundant populations. In the scRNA-seq data, CD8+ T cells from PI16/2 tumors showed type I IFN-activated and cytotoxic profiles, suggesting that partial SBRT (LDRT + HDRT) could trigger anti-tumor immunity. These transcriptomic observations are supported by the analysis of the effector markers IFNγ and Granzyme B in CD8+ T cells, showing an increase in the number of cells positive for these markers in partially irradiated tumors, in particular after anti-PD1 treatment. Although correlative, these data could suggest a role for activated CD8+ T cells in the anti-tumor response observed following this combined treatment, which was validated by depletion experiments. Of interest, CD8+ T cell depletion had a limited impact after total HDRT, in agreement with the view that the anti-tumor effect of ablative doses of RT is less dependent on immune activation than observed after PI treatments. Moreover, we also observed that CD8+ T cells from both portions of PI16/2 tumors were phenotypically less different than CD8+ T cells from distinct tumors totally irradiated with either 16 Gy or 2 Gy, as the CD8+ T cells from two volumes from PI16/2 tumors had few differentially-expressed genes. This strongly suggests a bilateral crosstalk between the adjacent parts of PI16/2. More generally, our results suggest that the immunological differences

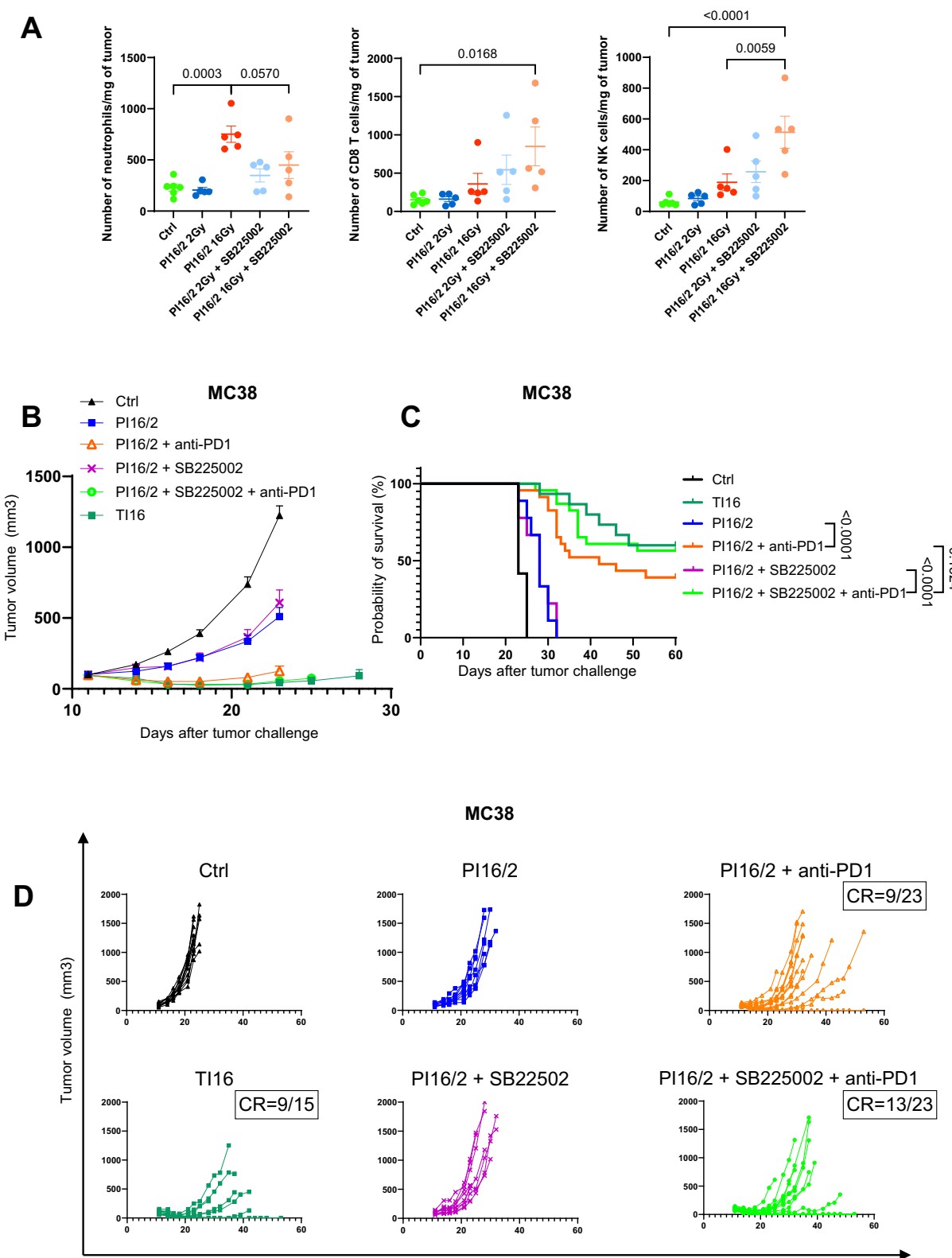

**A**

**B** MC38

**C** MC38

**D** MC38

observed in the 2 Gy region of PI16/2 tumors compared to the untreated regions in PI16/0 tumors could contribute to the different outcomes observed in Fig. 2A–C. These observations are supported by the fact that LDRT, when combined with HDRT and PD1 blockade, is able to induce a synergistic abscopal effect and effector CD8+ T cell infiltration in the TME in a model of subcutaneous MC38 implanted on both flank of mice, and had also shown efficacy in a clinical setting[18,29].

Hence, the immune effector response resulting from the interplay between HDRT and LDRT-treated areas and the synergy with PD1 blockade appear to be one of the armed wings of this combined treatment. In this regard, we found that the levels of the cytokinome in the tumor are differently modulated by the different irradiation modalities. Nevertheless, these analyses are limited to a single time point, and future studies will be needed to evaluate them with a longer

**Fig. 8 | CXCR2 blockade combined with anti-PD1 and partial irradiation with HDRT and LDRT improves tumor volume control by reducing the infiltration of neutrophils. A** Flow cytometry experiment analysis on neutrophils (left), CD8 T cells (center) and NKs (right) after treatment with SB225002. Subcutaneous MC38 tumors from C57BL/6 mice were treated and sampled as described in Fig. 2A, and SB225002 was administered on days 11 and 12. Data were represented in numbers of cells per mg of tumor. Differentially-treated parts of partially-irradiated tumors were separated and considered as different groups (Ctrl $n = 6$, $n = 5$ in all other treatment groups; from one experiment). Data are represented as the mean ± SEM, and $n$ represents the number of mice/groups. Numbers on these graphs represent $p$ values and were determined by ordinary one-way ANOVA with the Šidák's multiple comparison test. **B** Mean tumor volume growth of subcutaneous MC38 cells in C57/Bl6 mice, starting on the day of irradiation (Ctrl $n = 12$, TI16 $n = 15$, PI16/2 $n = 9$, PI16/2 + anti-PD1 $n = 23$, PI16/2 + SB225002 $n = 9$, PI16 + SB225002 + anti-PD1 $n = 23$; combined from five independent experiments). Curves are stopped as the first sacrifice occurs. Data are presented as mean ± SEM, and $n$ represents the number of mice/groups. **C** Kaplan−Meier survival curves of the efficacy experiment shown in (**B**). Numbers on the survival graph represent $p$ values and were determined by Log-rank Mantel−Cox analysis. **D** Individual curves of subcutaneous MC38 tumor growth in C57BL/6 mice (CR = complete response) of the efficacy experiment shown in (**B**, **C**). Source data are provided as a Source Data file.

kinetic, and to identify the cell types (tumor, immune, other stromal cells) producing such cytokines. Such studies could provide informative insights concerning the modulation of TIME induced by such a heterogeneous radiation dose distribution in the tumor.

We also observed a strong infiltration of neutrophils in the TME of PI16/2 tumors, as well as a marked shift in their phenotype. As a highly plastic type of cells, neutrophils are indeed known to be strongly influenced by their microenvironment and remain difficult to define since their function seems to evolve throughout tumor development[30]. However, this type of cells could also contribute to cancer progression through various tumor promoting functions including proliferation, aggressiveness, and dissemination, as well as immune suppression[31]. This ambiguous role has been illustrated by attempts to describe neutrophils subpopulations with a similar dichotomy to "M1-like"/"M2-like" macrophages; "N1-like" and "N2-like" respectively representing pro-tumor and anti-tumor phenotypes of neutrophils. However, key genetic markers describing these phenotypic states are yet to be fully identified and this oversimplified dichotomy mostly relies on functional and morphological differences[32]. Using scRNA-seq, we characterized neutrophils according to their differentiation level, as we identified "immature" features such as *SELL*, *RETNLG*, *HP* and *PGLYRP1*[33,34]. Pseudotime analysis was then used to associate neutrophil clusters with basic phenotypic characterization based on these settings. This classification was confirmed as the later stages of the pseudotime were represented by neutrophils expressing stronger levels of *PPIA*, *PTMA*, *MIF*, *CSTB*, and *PSAP*, which are commonly linked with more differentiated phenotypes[33,35–37]. These more differentiated phenotypes have been associated with an increase in neutrophil degranulation (release of cytotoxic molecules that trigger the apoptotic elimination of cancer cells) and inflammation activity, which can be described as anti-tumor features[33,38–40]. In addition to the increased infiltration of neutrophils in partially irradiated tumors combining LDRT and HDRT, scRNA-seq analysis revealed a phenotypic shift towards more "immatures" profiles. These profiles are often associated with immunosuppressive phenotypes, which was confirmed by the higher "T cell suppression score" and PD-L1 or CD206 neutrophil expression in these treatment conditions. These subpopulations of neutrophils are indeed able to dampen T cell activity through various mechanisms, such as abnormal vascularization, hypoxia, immune checkpoints or secretion of factors like Arginase 2, iNOS, MMP9 or IL10[41]. Germann and colleagues showed in immunostaining samples from colorectal cancer (CRC) human tumors that CD8+ T cells tended to be opposed to neutrophil localization, and to be mostly located at the border of neutrophil-infiltrated tumors[41]. According to their results, depletion of neutrophils could induce an increase in activated T-cell infiltration, as well as a trend to increased numbers of total T cells[41,42]. In our setting, neutrophils mainly infiltrate the 16 Gy portion (which is likely well controlled due to the ablative properties of HDRT) of the PI16/2 tumors at day 2 post-irradiation. However, considering the possible bilateral crosstalk between the two portions of these tumors, neutrophils from the regressing HDRT region could possibly traffic to the LDRT region and continue to play an immunosuppressive role. Different strategies have been explored to reduce the infiltration

of neutrophils in the TME, including targeting of the CXCR2 pathway. CXCR1/2 receptors promote neutrophil infiltration in the TME and are also thought to be a biomarker of immuno/radioresistance[43,44]. Accordingly, it has been shown that increased levels of IL-8, a human ligand of CXCR2, were associated with bad prognosis, confirming pre-existing data[25]. In our model, the CXCR2 receptor was more expressed in neutrophils from partially irradiated tumors combining LDRT and HDRT, and combining an anti-PD1 treatment to this irradiation scheme further increased the tumor levels of CXCR2+ neutrophils. Neutrophils expressing higher levels of CXCR2 are described in the literature as less "mature" as this feature is associated with a recent bone marrow outing, which was confirmed in our scRNA-seq analysis by the expression of CXCR2, mostly localized in "immature"/"intermediate" neutrophils[45]. Furthermore, studies have highlighted the potential benefits of inhibiting CXCR2-mediated neutrophil infiltration in the TME[41,45–50]. When we combined PI16/2 with SB225002, a potent, selective and non-peptide CXCR2 antagonist[51], we observed enhanced CD8+ T cells and NK cells infiltration in the TME in addition to the decrease in neutrophil infiltration, confirming the immunosuppressive properties of neutrophils recruited in a CXCR2-dependent manner, which favor immune exclusion[50,51]. These results contribute to explaining the increased efficacy of PI16/2 + anti-PD1 when combined with SB225002, associated with a trend for extended survival. Overall, these results suggest that CXCR2 inhibition may represent a promising approach to counteract the RT-induced immunosuppressive activity of neutrophils, even if further studies are needed to confirm the trends for a better anti-tumor response observed in the colorectal and breast cancer models. In line with this view, previous literature findings indicated that CXCR2 inhibitors can increase the efficacy of RT[52].

In a phase I clinical study combining pembrolizumab immunotherapy and multisite SBRT (delivered to at least two distinct metastases) in patients with advanced solid tumors, Luke et al. observed a good tumor control also in patients that could receive only partial tumor irradiations[24]. Recently, the same team published another article completing this study, showing that this partial irradiation setting was safe and well tolerated by patients, providing rationale for sparing organs at risk while administering a high dose to a partial tumor volume[28]. While constituting a promising clinical approach (especially in the context of oligometastatic, oligoprogressive diseases), the cellular and immune mechanisms underlying partial irradiation remained hardly explored, with only one preclinical attempt[22] and none using high-precision ballistic irradiation devices, and without tumor IR dose modulation. In view of our data, reducing the tissue volume receiving HDRT, and treating with LDRT the remaining tumor tissue, while combining with PD1 blockade appeared to be a promising strategy.

In our experimental work, CT imaging allowed the tumor to be precisely delineated in two portions of the same volume (50% of total tumor volume) for partial irradiation settings. These two hemisphere-shaped portions were then easy to separate for further analysis. We decided to study the immune characteristics of the differentially-treated extremities of partially irradiated tumors. For this purpose, we chose to separate tumors into three portions, as described in Methods. The two

extremities were used for analysis, while the intermediate part was discarded, to remove dose uncertainties considering the irradiated volumes. However, this "intermediate" part could also be interesting to study in order to understand the various immune interactions of high dose gradient areas. Such analyses using spatial transcriptomics will be carried out in future studies. These further analyses may also contribute to a better comprehension of the immune changes undergoing in tumors when SFRT techniques (i.e., GRID/lattice, minibeam and microbeam) are used, as they deliver alternance of high doses (peaks) and low doses (valleys), with several transition areas[53]. Additionally, it would be conceivable to imagine that toxicity could decrease in PI settings compared to total volume HDRT, as it is observed in the frame of SFRT[54]. Using PI in the clinics would allow to only target tumor areas that are distant from the potential OARs with HDRT, and to target the rest of the tumor (which is closer to OARs) with LDRT, therefore limiting toxicity to said organs. Further studies will be performed in order to assess the reduction of side effects of PI compared to TI in preclinical settings using orthotopic tumors, which are more suitable to perform toxicity studies than subcutaneous ones.

It seems relevant to consider optimizing the administration of the irradiation dose. Considering the recent progress in imaging, but also in radiomics, it seems now possible to take tumor heterogeneity (e.g., hypoxia, tumor areas that are poorly infiltrated with T cells, etc.) into account for the identification of the radiation dose to administer to the different tumor areas in various SFRT settings[23,55–57]. An approach involving the targeting of hypoxic areas has been proposed and tested in clinical settings (PArtial Tumor irradiation targeting HYpoxic segment, PATHY)[23,58]. Moreover, instead of highly different IR doses as used here as a model to obtain proof of concept and to study the interplay between differently irradiated volumes, gradients may be applied and would likely result in improved efficacy. Dose fractionation is another parameter to be considered. In this paper, we administered RT with a single dose, as it was technically easier with the use of image-guided SBRT. Since it has been shown that fractionated RT could be more likely to synergize with ICI than single-dose RT, and induce ICD[59–62], future attempts involving fractionated partial RT may result in further improved efficacies.

In conclusion, we showed that partial irradiation combining LDRT + HDRT, associated with the blockade of PD1 and CXCR2 was a potent way to promote anti-tumor activity, leading to several complete responses. While further studies are needed to improve and optimize partial irradiation settings, our work may open new fields to both reduce RT toxicity and to improve therapeutic index of RT and ICI combinations. Our results have translational relevance, especially considering that CXCR2 inhibitors are already used in the clinics[63,64]. This approach may be tested first in clinical contexts where IR of whole tumor volume is not feasible, i.e., when tumor masses are too large, or close to radiosensitive organs, preventing the tumoricidal high doses to the whole tumor target.

## Methods

### Study approval
Animal experiments were performed in compliance with French and European regulations on the protection of animals used for scientific purposes (EC Directive 2010/63/EU and French Decree 2013–118). All experiments were approved by the Ethics Committee CEEA26 of Gustave Roussy (approval number I-94-076-11) and #81 at IRSN (approval number E92-032-01) and authorized by the French Ministry of Research.

### Cells
MC38 colon cancer cells were purchased from Kerafast. Cells were cultured with Dulbecco's Modified Eagle Medium (DMEM, Gibco) adjusted to contain 10% fetal bovine serum (FBS), 1% penicillin/streptomycin, 1% sodium pyruvate, 1% HEPES buffer solution, 1% non-essential amino acids (NEAA). CT26.WT (hereafter named CT26) colon carcinoma and 4T1 mammary carcinoma cells were purchased from ATCC. Both tumor cell lines were cultured with RPMI-1640 medium (Gibco) adjusted to contain 10% FBS and 1% penicillin/streptomycin. Cells were cultured in humidified incubators at 37 °C with 5% carbon dioxide.

### Animal experiments
C57BL/6 and BALB/c female mice aged 7–8 weeks were purchased from Janvier CERT (Le Genest St. Isle, France), and housed at the Gustave Roussy animal facility (Plateforme d'Evaluation Preclinique, PFEP) or at the IRSN Animal Research and Ethics Support Group (GSEA). All animals were included in experiments after at least 1 week of acclimatization period. $10^6$ cells resuspended in 50 μL of phosphate-buffered saline (PBS) were subcutaneously injected into the right hind flank of each mouse. After 8 (CT26 model) or 11 (MC38 model) days, when tumors reached 140–150 (CT26) or 100–130 (MC38) mm³, the mice were randomly divided into treatment groups. $0.75.10^6$ 4T1 cells resuspended in 50 μL of PBS were orthotopically injected in the fat pad of each BALB/c mouse. After 13 days, when 4T1 tumors reached around 100 mm³, the mice were randomly divided into treatment groups. Measures were performed with a caliper, and tumor volume was calculated as follows: (width² × length) × 0.5.

All animal experimental procedures were carried out in accordance with the Ministry of Higher Education, Research and Innovation (MESRI) regulations with specific authorizations. Mice were maintained in specific pathogen-free facilities. They had free access to food and water. They were housed on a 12-h light/dark cycle at a room temperature of 22 °C ± 2 °C and a relative humidity of 55% ± 15% (five animals per cage, disposable ventilated cages, Innovive, France). The health and behavior of the mice were assessed three times per week. Mice were euthanized by cervical dislocation on the presentation of defined criteria (tumor size exceeding 1500 mm³, advanced tumor necrosis, global behavior), and a survival time was recorded to perform a survival analysis for the treatment groups. In some cases, the size limit has been exceeded the last day of measurement and the mice were immediately euthanized. We adhered to criteria and guidelines in all experiments.

### Antibodies and treatments
The anti-PD1 (clone RMP1-14, BE0146), and the rat IgG2a isotype control (clone 2A3, BE0089) were purchased from BioXcell, and administered i.p. at 10 mg/kg after SBRT and at 5 mg/kg 3/week for 2 weeks. For depletion of CD8+ T cells and neutrophils, the anti-CD8α antibodies (5 mg/kg, BioXcell, clone 2.43, BE0061) and anti-Ly6G (10 mg/kg, BioXcell, clone 1A8, BE0075-1), respectively, were injected i.p. on days 0, 4, and 7 post SBRT. Rat IgG2b (5 mg/kg, BioXcell, clone LTF-2, BE0090) and rat IgG2a isotype control (10 mg/kg, BioXcell, clone 2A3, BE0089), respectively, were used as controls. CXCR2 inhibitor SB225002 (10 mg/kg, MedChemExpress, HY-16711) was dissolved in a 1% dimethyl sulfoxide (DMSO), 20% polyethylene glycol 400 (PEG400), 5% polysorbate 80 (Tween 80), and 74% ddH2O[51]. SB225002 was administered i.p. at 10 mg/kg after SBRT, and every day for 8 days.

### Irradiation
Before irradiation, mice were anesthetized with 100 mg/kg ketamine (Imalgene 1000, Merial, France) and xylazine (Rompoun 2%, Bayer Healthcare, France) or 2.5% isoflurane. Before and after the procedure, anesthetized mice were put in a Thermacage (Datesand). Two Small Animal Radiation Research Platforms (SARRP, XStrahl), microirradiation systems guided by cone beam computed tomography (CBCT) images, were used for irradiation of mice. CBCT images were obtained using an uncollimated beam (20 × 20 cm), a voltage of 60 kV, a current of 0.8 mA with inherent and additional filtrations of 0.8, and 1 mm of beryllium and aluminum, respectively, with continuous beam on and

360 stage rotation between the x-ray source and the digital flat panel detector. The 3-dimensional (3D) reconstruction images and dose planning were performed with the Muriplan treatment planning system (TPS)[65]. Irradiations were performed at a voltage of 220 kV and a current of 13 mA with inherent and additional filtration of 0.8 of beryllium and 0.15 mm (with an additional filtration of 0.105 mm on the second machine) of copper. The protocol developed by the Radiation Therapy Committee Task Group 61 and presented by the American Association of Physicists in Medicine was followed as much as possible in order to evaluate the half value layer (0.667 mm and 0.844 mm, copper) inside of the two SARRP[66]. The configuration used to obtain the half value layer has been previously described[67]. Two treatment plans were designed to perform partial tumor irradiations (50 % of the tumor receiving 2 Gy, and 50 % of the tumor receiving 16 Gy): first, a dose of 2 Gy was delivered in a single fraction to the whole tumor volume, using two beams which were at 180° to each other to ensure homogeneous dose distribution inside the tumor. A dose of 14 Gy was then administered to 50% of the tumor volume using tumor segmentation and a dose-volume histogram (DVH). For whole tumor volume irradiation, the isocenter was placed on the tumor, and a two-beam treatment was planned to deliver the prescribed total dose to the isocenter. For partial tumor volume irradiation, the isocenter was placed to deliver 16 Gy to 50% of the tumor volume with a two-beam treatment. In function of the size and shape of the tumor, collimators of 3 × 3, 5 × 5, or 10 × 10 mm² were used. Both beams were at 180° to each other to ensure homogeneous dose distribution inside the tumor. The two halves of the tumor were separated following a transverse section. Examples of TPS and associated DVH are shown in Fig. 1B, C, respectively.

## Tumor sampling

To collect tumor samples, preserving their spatial orientation to physically recognize and separate the 16 Gy-irradiated and non-irradiated (or 2 Gy-irradiated)-portions, an ink staining procedure was developed. Briefly, two inks with different colors were used to stain the differentially irradiated parts of the tumor when collecting it after mouse euthanasia. This allowed an easy orientation of the tumors for histological analysis. A schematic figure of tumor sampling and orientation is shown in Fig. 1D. To have a clear separation of the differentially irradiated volumes for flow cytometry/protein/Single-cell RNAseq analysis, an intermediate portion was removed at the interface of both areas after ink staining, following the same transverse section. The two other volumes (corresponding to differentially irradiated tumor volumes) were then put in distinct collection tubes.

## Flow cytometry analysis

C57BL/6 mice were euthanized, and MC38 tumors were collected as described in "Tumor sampling" section. Tumors were weighed, finely chopped, and dissociated with digestive enzyme (Tumor Dissociation Kit, mouse, Miltenyi Biotec, 130-096-730) according to the manufacturer's instructions for 30 min at 37 °C in the ThermoMixer® C (Eppendorf). Tumor digestions were then mechanically disrupted and filtered through a 40 μm nylon filtration cell strainer (Cell Strainer 40 μm Nylon, Falcon, USA, 352340).

Resulting single cells were incubated at 4 °C with anti-CD16/32 (clone 93, BioLegend, 101319, 1/100) antibodies for 10 min. The cells were stained using the same protocols as previously described[68,69], with the following antibodies: anti-CD69 APC (clone H1.2F3, BD Biosciences, 560689, 1/200), anti-CD11b BUV395 (clone M1/70, BD Biosciences, 563553, 1/400), anti-CD8a BV421 (clone 53−6.7, BD Biosciences, 563898, 1/400), anti-CD62L BUV737 (clone MEL-14, BD Biosciences, 612833, 1/200), anti-CD45 PerCP-Cy5.5 (clone 30-F11, BD Biosciences, 550994, 1/200), Anti-CD45 FITC (clone 30-F11, BioLegend, 103107, 1/200), anti-CD45 APC-Cy7 (clone REA737, Miltenyi Biotec, 130-110-662, 1/50), anti-Siglec F PE-CF594 (clone E50-2440, BD

Biosciences, 562757, 1/100), anti-LAG3 BV650 (clone C9B7W, BioLegend, 125227, 1/50), anti-PD1 (CD279) BV605 (clone 29F.1A12, BioLegend, 135220, 1/100), anti-CD25 PE-Cy7 (clone PC61, BD Biosciences, 552880, 1/100), anti-CD4 BV510 (clone RM4-5, BioLegend, 100559, 1/400), anti-CD19 PE-CF594 (clone 1D3, BD Biosciences, 562291, 1/200), anti-NK1.1 FITC (clone PK136, BD Biosciences, 553164, 1/100), anti-Ly6G BV421 (clone 1A8, BD Biosciences, 562737, 1/400), anti-Ly6G PerCP Cy5.5 (clone REA526, Miltenyi Biotec, 130-117-500, 1/50), anti-I-A/I-E BV510 (clone 2G9, BD Biosciences, 743871, 1/200), anti-CD64 PE-Cy7 (clone X54-5/7.1, BioLegend, 139314, 1/200), anti-CD206 PerCP Cy5.5 (clone C068C2, BioLegend, 141716, 1/200), anti-CD274 BV650 (clone MIH5, BD Biosciences, 740614, 1/100), anti-Ly6C APC-Cy7 (clone AL-21, BD Biosciences, 560596, 1/100), anti-Ly6C AlexaFluor 700 (clone HK1.4, BioLegend, 128024, 1/50), anti-CD11c BV605 (clone N418, BioLegend, 117334, 1/100), anti-CD103 BV711 (clone 2E7, BioLegend, 121435, 1/100), anti-Gr1 PE (clone RB6-8C5, Thermo Fisher Scientific, 14-5931-82, 1/200), and anti-CD182 (CXCR2) PE (clone 3F10-B3, BioLegend, 163903, 1/50). To analyze IFNγ and Granzyme-B intracellular levels, cells were stimulated for 2 h at 37 °C with PMA/ionomycin before incubation with anti-CD16/32 antibodies followed by membrane staining. Cells were then fixed using 4% paraformaldehyde for 15 min at 4 °C and permeabilized using Perm/Wash Buffer (BD Perm/Wash) for intracellular cytokine staining. Anti-IFNγ BV786 (clone XMG1.2, BD Horizon, 563773, 1/50) and anti-Granzyme B PE-Cy7 (clone NGZB, eBioscience, 25-8898-80, 1/200) were used for intracellular cytokine staining. All events were acquired immediately following sample processing using an LSR Fortessa flow cytometer (BD). All flow cytometric data analysis was performed using FlowJo v10.8.1 (FlowJo, Ashland, OR, USA). The myeloid cell gating strategy is the same as previously described[68,69], and the lymphoid cell gating strategy is described in Supplementary Fig. 2A.

## Cytokine/chemokine array

MC38 tumor cells were collected, and partially irradiated tumors were separated into two distinct groups (one for each treatment condition), following the method described in "Tumor Sampling". Tumors were weighed and mechanically disrupted in RIPA lysis buffer (150 mM NaCl, 1% NP-40, 0.25% Na-deoxycholate, 50 mM Tris-HCl (pH 7.4), Sigma-Aldrich, USA) containing a protease inhibitor cocktail (Roche, Switzerland, 04-693-124-001) using Biomasher disposable homogenizers (Nippi, Japan, 320102). The cytokine and chemokine concentrations in tumor tissues were profiled using the Mouse Cytokine/Chemokine 44-Plex Discovery Assay® Array (MD44) at Eve Technologies Corporation (AB, Canada). Protein extracts were diluted to 4 μg/μL, and the multiplex immunoassay was analyzed with the BioPlex 200 instrument (Bio-Rad, USA). Cytokine and chemokine concentrations were calculated based on the standard curve generated using the standards included in the kit. The concentrations of the following cytokines were calculated: Eotaxin, G-CSF, GM-CSF, IFNg, IL-1a, IL-1b, IL-4, IL-5, IL-6, IL-7, IL-10, IL-12p40, IL-12p70, IL-13, IL-15, IL-17, IP-10, KC, LIF, LIX (CXCL5), MCP-1, M-CSF, MIG, MIP-1a, MIP-1b (CCL3), MIP-2, RANTES (CCL5), TNFa, VEGF, Fractalkine, IFNb-1, IL-11, MDC, MIP-3a and TARC.

A Partial Least-Squares Discriminant Analysis (sPLS-DA) was conducted on the 35 cytokines within the dynamic range of the assay, as a supervised multivariate analysis to discriminate the different treatment groups based on their cytokine/chemokine concentrations. More precisely, the sPLS-DA maximizes the covariance between the cytokine/chemokine covariates by constructing components (linear combinations of the original covariates of each block) that are maximally correlated with respect to an outcome variable, here the treatments groups. In addition, the lasso penalization was included in the loss function of the model to ensure sparsity, variable selection, and interpretability of the sPLS-DA components[70]. In all the analyses, leave-one-out cross validation was used to determine the model parameters including number of components and number of features per

component. All the sPLS-DA study was performed using the package "mixOmics" (version 6.18.1) in R software.

### IHC and histological analysis

MC38 tumor cells were collected (15 min after RT for gH2AX staining) and spatial orientation of partially irradiated tumors was indicated with the ink staining procedure described in "Tumor Sampling". Tumors were kept in a 4% paraformaldehyde (PFA) solution for 24 h, paraffin embedded, and then cut into 4 µm sections. γH2AX (ser139) immunostaining (Abcam, Ab81299) was performed using a Ventana Benchmark automaton (Ventana, Arizona, USA), and digitized using a slide scanner (NanoZoomer S60, Hamamatsu Photonics, France).

### Single-cell RNA sequencing and analysis

$10^6$ MC38 cells resuspended in 50 µL of phosphate-buffered saline (PBS) were subcutaneously injected into the right hind flank of each C57BL/6 mouse. After 11 days, when tumors reached around 100–130 mm³, the mice were randomly divided into treatment groups. Mice were then treated according to their group and 2 days after RT, mice were euthanized, and tumor tissues collected as described above. Single-cell suspensions were prepared and CD45$^+$ cells were positively selected using CD45 MicroBeads (Miltenyi Biotec, 130-052-301). Viability was measured using ViaStain AOPI Staining Solution (Ozyme, CS2-0106-5ML), and cell concentration and the % CD45-positive cells were evaluated by flow cytometry using Fixable Viability Dye eFluor™ 780 (Thermo Fisher Scientific, 65-0865-14) and CD45 FITC (clone 30-F11, BioLegend, 103107). At least 85% of cell viability and >90% of CD45$^+$ cell was obtained for further r encapsulation. For each sample, 10,000 cells were encapsulated using the 10x Chromium Controller system and a Chromium Next GEM Single Cell 3′ Dual Index Kit v3.1. Reverse transcription and libraries were prepared according to the manufacturer's instruction. Two independent experiments were performed including 7 experimental conditions. In total, 16 samples were included in the study (4 replicates of unirradiated controls, 2 replicates for the other conditions). Library quantification and quality was performed using the Agilent Bioanalyzer 2100 (Agilent Genomics). Indexed libraries were equimolary pooled and sequenced on an Illumina NovaSEq 6000 sequencing system with a minimum depth sequencing of 20,000 read pairs/cell. Single-Cell RNA-Seq outputs were processed using the Cell Ranger software (10x Genomics). Downstream analysis was performed using the Seurat 4.0 and Monocle 3 packages in R.

Seurat objects corresponding to each sample were generated separately using the aggregated filtered barcode matrix files (Cell Ranger output), and initial quality control steps were performed to remove low-quality cells and doublets. Cells with >10% mitochondrial reads, <200 or >3500 expressed genes, and <1000 reads were removed from the Seurat objects. Each Seurat object was identified with the associated treatment group. Data normalization through the NormalizeData function was then processed for each Seurat object with the LogNormalize normalization method. Biologically relevant variable genes were then identified with the FindVariableFeatures function for each Seurat object, using the vst selection method (nfeatures = 2000). Selected features from each Seurat object were then scaled and centered using the ScaleData function. For each treatment condition, replicate samples were then merged together. Normalization and scaling were then applied to the Seurat objects representing each treatment condition with the same parameters as previously described. Seurat objects were then randomly downsampled with a threshold of 15,066 cells for each treatment condition. A global Seurat object was then generated, normalized, and scaled, combining all treatment groups. Using the 2000 variable genes thus identified as input, a principal-component analysis (PCA) was performed, and the top 30 principal components were retained for further Uniform manifold approximation and projection (UMAP) visualizations.

Subsequent cell clustering using the Louvain algorithm and visualization of the cells grouped by treatment condition were performed. We used conventional functions in Seurat FindMarkers and FeaturePlot to identify and locate differentially-expressed genes (DEGs). These DEGs were determined by a nonparametric Wilcoxon test, and ranked based on the avglog2(FC), with a cutoff of adjusted $p$ value < 0.05. The different cell types were identified within the Seurat clusters/groups of clusters, according to the literature and the DEGs. In order to identify subpopulations among different cell types, associated clusters were isolated and integrated into a new Seurat object. After data normalization and scaling, a PCA was performed with the previously described settings. Neutrophils and CD8$^+$ T cells UMAPs were respectively generated using a Louvain resolution of 0.2 and 0.3. Subpopulations were identified using specific markers and literature, as indicated in Supplementary Figs. 3 and 4[33,71,72]. Heatmaps were generated using the DoHeatmap function, with a downsample $n = 100$ and specific functional scores were calculated using the AddModuleScore function, which calculates module scores for feature expression programs. AddModuleScore results were shown using the function ggplot, associated with the functions geom_boxplot and geom_violin. Libraries from the Gene Ontology platform were used to generate the different module scores (GO:0001916 for the "positive regulation of T-cell mediated cytotoxicity" score, GO:0060340 for the "positive regulation of type I interferon-mediated signaling pathway", GO:0043315 for the "positive regulation of neutrophil degranulation"). "T cell suppression" score was generated using key markers of T cell suppression in neutrophils[41]. To visualize DEGs between treatment conditions, volcano plots were generated using the ggplot and geom_point functions. Genes with >0.4 avglog2(FC) and an adjusted $p$ value of 0.05 are labeled and highlighted in red (upregulated) or blue (downregulated).

To generate Pseudotimes, neutrophils and CD8$^+$ T cells were isolated from the global Seurat object into independent Seurat objects as described above, and were injected into Monocle3 package[73,74]. Cells were clustered following Leiden community detection using cluster_cells function, with k = 50 nearest neighbors used to create clustering graphs. Gene expression trajectories were learnt with the learn_graph function, with a maximum number of nearest neighbors to compute in the reversed graph embedding = nn.k = 50. The function graph_test was then used, in order to identify differentially-expressed genes across the trajectory, sorted by their Moran's I. The expression family function used for the test was "negbinomial", and 4 cores were used while testing each gene for differential expression.

For specific supervised analysis of neutrophils, a UMAP was performed to improve readability and interpretability of the data[75]. UMAP is a nonlinear dimension reduction technique considered now as a gold standard for dimension reduction of single cell data. While UMAP is often used in literature in its unsupervised formulation, its algorithm offers significant flexibility allowing it to be extended to include categorical label information to do supervised dimension reduction. This supervised UMAP appears particularly appealing here to enhance the structure of the embeddings according to the numerous different types of RT treatments. This ensures a better separation between the different labels while maintaining close those producing similar transcriptomic profiles. The supervised UMAP was performed using the UWOT (version 0.1.11) R software package[76].

### ELISA

MC38 tumor cells were collected, and partially irradiated tumors were separated into two distinct groups (one for each treatment condition), following the method described in "Tumor Sampling". H2AX phosphorylation was evaluated using the PathScan® Phospho-Histone H2A.X (Ser139) Sandwich ELISA Kit (Cell Signaling Technology, 50929C).

### Statistical analyses

Data are represented as the mean ± standard error of the mean (s.e.m.) or standard deviation (s.d.) as indicated in the figure legends. Normality was evaluated by the Shapiro–Wilk test. When more than two

groups with gaussian distribution were compared, one-way ANOVA with Sidak's multiple comparisons was used or Kruskal–Wallis with Dunn's test for multiple comparisons as indicated in the figure legends. ROUT test with $Q = 2\%$ was used to identify outliers. Two-way ANOVA was used for γH2AX staining analysis. Log-rank Mantel–Cox analyses were performed for survival curve comparisons, with no correction made for multiple comparisons. All statistical tests were performed using GraphPad Prism software v.10 (GraphPad Software). In all types of statistical analysis values of $p < 0.05$ were considered significant ($*p < 0.05$; $**p < 0.01$; $***p < 0.001$; $****p < 0.0001$).

### Reporting summary
Further information on research design is available in the Nature Portfolio Reporting Summary linked to this article.

## Data availability
The scRNA-seq datasets generated in this study have been deposited in the Gene Expression Omnibus (GEO) under accession code GSE262699. The remaining data are available within the Article, Supplementary Information or Source Data file. Source data are provided with this paper.

## Code availability
All analytical codes to reproduce scRNA-seq figures are available at https://doi.org/10.5281/zenodo.13684101. All software used in this study are publicly available and are described in the "Methods" section.

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

## Acknowledgements

This work has received financial support Canceropole IdF (projet Emergence) to M.M., INSERM and SIRIC SOCRATE to E.D., the Fondation ARC pour la Recherche sur le Cancer (projet fondation ARC) to M.M., the French National Research Agency within the FRANCE2030 investment plan (grant application No. ANR-21-RHU5-0005) to E.D., The French National Cancer Institute (INCa) under the AAP SEQ-RTH22, project INCa_16861 to F.M. and M.M.; and under the AAP PLBIO-2024, projet INCa_19441 to F.M. and M.M. We thank the staff of the animal facilities at IRSN and GR for invaluable expertise, P. Rameau, and C. Catelain at the PFIC platform at Gustave Roussy, and P. Gonin and K. Ser-Le-Roux at the PFEP platform at Gustave Roussy.

## Author contributions

Study design: F.M., E.D., and M.M. Methodology, acquisition of data and analysis of data: P.B., M.D.S., L.S., G.T., J.L., W.L., M.G.d.T., C.C., L.M., C.S., G.M., K.B., M.A.B., F.M., and M.M.; writing of the manuscript: P.B. and M.M.; review and edition of the manuscript: P.B., M.D.S., L.S., G.T., J.L., W.L., M.G.d.T., C.C., L.M., C.S., G.M., K.B., M.A.B., F.M., E.D., and M.M.

## Competing interests

P.B., L.S., W.L., M.G.D.T., C.C., L.M., C.S., G.M., E.D. and M.M. declare grants from Roche Genentech, AstraZeneca, Merck Serono, Bristol-Myers Squibb, Boehringer Ingelheim, Eli Lilly and MSD, outside the submitted work. E.D. declares consulting fees from Graegis. The remaining authors declare no competing interests.
