## [Peer Review File · Nature Communications]

Non-homogenous intratumor ionizing radiation doses synergize with PD1 and CXCR2 blockadeEditorial note: Parts of this Peer Review File have been redacted as indicated to maintain the confidentiality of unpublished data.

REVIEWER COMMENTS

Reviewer #1 (Remarks to the Author):

Even though being ignored for decades, it has become obvious that targeted radiotherapy (RT), in addition to the frequently mentioned immune-suppressing effects, has also some immune-active ones. Currently, a paradigm shift takes place that suggests more individual radiation applications regarding dose and volume of irradiation when considering, besides the local effects of RT on tumor cell clonogenicity, also the local and systemic immune modulatory consequences. Most likely RT schemes will have to be adapted in the future and more individualized concepts will be needed.

The paper of Bergeron et al. picks up on one of these possible very promising innovative approaches. They hypothesized that making use of both, killing of tumor cells by high dose RT in combination with immune stimulating effects of lower doses of RT should be beneficial for induction of anti-tumor immune responses, at least with additional immune modulation. Regarding the latter, the approach to test inhibition of the PD-1/PD-L1 axis is well comprehensible, as RT has shown in various settings to increase expression of the immune suppressive checkpoint molecule PD-L1 and partly also PD-L2 on tumor cells. They focus on colorectal cancer as model system (MC38 tumors ectopic in C57BL/6 mice) and focus on irradiation parts of one tumor mass with both, 2Gy and 16Gy RT. This requires a very precise irradiation of parts of the tumor and could be achieved clinically with SBRT.

In their preclinical settings it was achieved to perform irradiations with millimetric precision to deliver different irradiation doses. They found that irradiation of one tumor mass with both low and high dose RT resulted in infiltration of CD8+ T cells, what is generally known, but also of neutrophils, being a new discovery. Neutrophil count increase is known as negative predictor for immune therapies of solid tumors and the new finding of Bergeron and colleagues shed light into possible mechanisms: CXCR2 was identified as a target to reduce

neutrophil infiltration. In addition, single-cell RNA-seq revealed that this new irradiation setting reshapes the tumor-infiltrating CD8+ T cells into a more cytotoxic phenotype. At least in their model system a multimodal therapy of partial irradiation with low and high doses in combination with inhibition of PD-1 and CXCR2 was effective in inducing anti-tumor immunity.

To further stress the new findings the authors should consider the following:

Please already define in the abstract why a single dose of 2Gy is a “low dose”, since this dose is the standard single dose of “high dose RT” for solid tumors

What was the rationale for choosing 16Gy as high dose?

In the introduction LDRT has to be defined more detailed, as the term LDRT is also used for irradiation of chronic inflammatory and degenerative diseases with as single dose per fraction of 0.5Gy. This should be discussed and respective literature be cited.

It would have been nice to validate the results in an orthotopic setting. However, it is clear that in a first setting focus has to be set on precise irradiation and that's why the ectopic approach might be more elegant. The data in Figure1 even prove biologically with γ H2AX staining that one part of the tumor was exposed with a high dose and the other part with a low dose, confirming the physical irradiation planning and setting. Nevertheless, this should at least be discussed in more detail.

Data of Figure 2 indicate that RT with 16Gy to only one part of the tumor in combination with 2Gy and anti-PD1 is as effective as RT with 16Gy to the whole tumor. It would be nice to see as control also data of the setting of irradiation of the whole tumor with 2Gy alone and in combination with anti-PD-1.

The following mechanistic analyses indicate that partial irradiations with 16Gy and 2Gy create a distinct immune micro-environment being mainly characterized by infiltration of CD8+ T cells, NK cells and neutrophils with an immune suppressive phenotype. It was

confirmed by RNA-seq analyses and mainly CD8+ T cells are phenotypically re-shaped following partial irradiation. Analyses of the neutrophil cluster revealed that these cells could be distinguished as three distinct groups. These results are the basis to optimize multimodal settings by specifically targeting these immune cells. Depletion of neutrophils improved tumor control and mouse survival when combined to partial irradiations of 2Gy and 16Gy and anti-PD1. Further, using a selective CXCR2 antagonist revealed that it prevented the neutrophil increase. The advantage in comparison to whole irradiation with 16Gy might be less toxicity. However, data about toxicity are missing and should be included or at least being discussed in detail.

Are the observed effects general ones or just being observed in the MC38 model? A second model for confirmation would be very valuable.

Reviewer #2 (Remarks to the Author):

In this study, Bergeron and colleagues use precise irradiation (IR) techniques to gain novel insights into the modulation of the tumor microenvironment (TME) in response to non-uniform radiation doses. Specifically, subcutaneously-implanted colorectal cancer-derived MC38 cells, upon tumor growth, received different IR regimens. Applying high dose (16Gy) on one tumor half and low dose (2Gy) (PI16/2) on the other revealed distinct TME characteristics compared to uniformly irradiated tumors, affecting cytokines, immune cell abundance and phenotype. Importantly, PI16/2 synergized with anti-PD1 treatment with even some complete responses observed, while each treatment modality (anti-PD1 or PI16/2) alone only modestly diminished tumor growth and augmented mouse survival. The study also explored the molecular alterations and importance of neutrophils, notably by preventing their recruitment through CXCR2 blockade, which synergises with anti-PD1 upon concomitant PI16/2 IR. Overall, the approaches undertaken and data obtained are fundamentally important; the methodology is sound; the data show promise in enhancing the efficacy of radiation and immunotherapy combinations, particularly in cases where whole-tumor irradiation is impractical due to tumor size or proximity to sensitive organs. The findings, in addition to their fundamental insights, could thus have strong clinical value.

I have a few comments, which I hope will be helpful to the authors.

Major comments:

1. Although the work done is original and impressive, a potential weakness is its reliance on a single cell line model grown subcutaneously. It would be nice to be able to confirm at least some findings using another tumor model with similar IR (PI16/2) procedures.

2. It is important to validate neutrophil depletion appropriately, because anti-Ly6G often fails to deplete these cells, depending on the protocol used, mouse age + background, and because many reports from the literature use similar antibodies for depletion and for validation, without considering antigen masking. Here, how was neutrophil depletion validated (Fig. S4A)? The authors should be careful about this point, especially as anti-Ly6G induces neutrophil turnover, so younger cells could be released from the bone marrow and influence the phenotype observed. Some useful information may be found in Nature Com 2020 (PMID: 32488020).

3. Linked to Figure 5: CD8 T cell phenotypes are only inferred from transcriptional signatures. It would be interesting to test or validate some changes at the protein level, to increase confidence about their effector functions. Specifically, do CD8 cells upon PI16/2 IR express different levels of cytoplasmic IFN γ or TNF upon in vitro reactivation (PMA+ionomycin)?

Minor comments:

1. Validate CD8 T cell depletion efficacy.

2. The authors mention that the expression of 44 cytokines was tested. It seems some information is lacking as of the technique used and the identity of the other cytokines. Were they undetected, or not significantly changed?

3. Linked to Figure 3D, it would be interesting to identify which cell types produce the differentially expressed cytokines. Are these tumor cells, or immune cells, perhaps directly

neutrophils? Also, was Cxcl5 expressed and induced by PI? Finally, kinetics of cytokine measurements and neutrophil phenotypic analyses would be interesting, but perhaps for future follow-up studies.

4. Figure 3B: reviewer wonders if the scale “fold increase” should not be called instead “fold change”, as decreases are also observed in comparison to control. Also, is it correct to start at “0”, while a fold increase of 0.5 should equal to -2, and as, with this scale, a decrease of 2-fold appears smaller, visually, to an increase of 2-fold.

5. Figures 2CD+6AB+S4BC: statistics are missing.

6. Legend of Figure 1: Replace ‘D’ by ‘F’.

7. Line 231, reviewer is unsure of the meaning “Corresponding to different degrees of plasticity”, when referring to the three neutrophil clusters.

Reviewer #3 (Remarks to the Author):

Bergeron et al

NCOMMS-23-48193

Non-homogenous intratumor ionizing radiation doses synergize with PD1 and CXCR2 blockade

Summary

The manuscript describes the impact of reducing the tumor area that is irradiated with a higher dose and adding low dose radiation to the remaining tumor. The results nicely show that while a total high dose of 16Gy is effective, only irradiating half of the tumor with this dose results in a loss of tumor control. Adding anti-PD1 to this partial treatment is poorly effective. Irradiating with a combination of 16Gy and 2Gy results in improved control over 16Gy combined with no dose when combined with anti-PD1 therapy.

The authors see a clinical rationale for reduced dose, where there are limited opportunities

for high doses to all of the tumor due to risk of toxicities. However, a split dose is less effective than a total high dose in this tumor, so while they show that giving some irradiation to the remainder is better than nothing, it does not provide evidence that split doses are optimal. In fact, the 16/2Gy treatment plus anti-PD1 remains less effective than 16Gy to all of the tumor without anti-PD1, and in this MC38 model doses in the 16Gy range plus anti-PD1 are generally curative. Nevertheless, the data does clearly demonstrate a value to adding some dose over no dose to the parts of the tumor that do not receive 16Gy.

Analysis of the tumor with flow cytometry and with scRNASeq show some T cell and neutrophil correlates with the 16Gy/2Gy response, but these are not very consistent. CD8 T cells are shown to be necessary for tumor control by RT plus anti-PD1, which is expected. Neutrophil increases are seen in the 16Gy/2Gy treatment and not seen in the 16Gy/0Gy therapy, and postulated as a limitation in tumor control. However, neutrophils also increase in 16Gy total dose, which is effective. In addition, it is not clear why anti-PD1 is not included in these studies. 16Gy/2Gy is very similar in tumor control to 16Gy/0Gy in the absence of anti-PD1, so the neutrophil change cannot be postulated as a reason for the different outcome – there is no different outcome in the samples where neutrophils change.

The manuscript is well written, the figures are clear and the conclusions are generally valid. However, the limitations discussed above and below reduce the overall impact of the work.

Major issues.

This is a single tumor model. It is not reasonable to infer too much from these results in MC28 colorectal carcinoma unless they are validated in additional tumor models.

The CXCR2 inhibition does not improve tumor responses when combined with 16Gy/2Gy treatment; it only improves responses when also combined with anti-PD1. The changes in neutrophil differentiation associated with 16Gy/2Gy treatment may have nothing to do with the mechanism, as we do not know what happens when anti-PD1 is also present, where the different response is seen. Since 16Gy/2Gy treatment changes neutrophil maturation and yet CXCR2 inhibition does not improve tumor responses unless anti-PD1 is present, the proposed mechanism is not proven. The authors need to model the response – show what is

different when anti-PD1 is combined with 16Gy/2Gy versus 16Gy/0Gy. Clearly, CXCR2 inhibition has an effect here, but is this actually correlated with the maturity effect observed without anti-PD1?

The observation that CD8 T cell depletion eliminates tumor control by 16Gy/2Gy combined with anti-PD1 is not sufficient to validate the authors' conclusions on CD8 T cell differentiation in the tumor. That anti-PD1 functions via CD8 T cells is evident and very well demonstrated in this and other models. If the authors aim to link the changes in T cell differentiation in the tumor to treatment responses, then more targeted interventions are needed to determine whether these slightly different populations are important. Otherwise these observations that T cell differentiation is slightly different remain correlations and not mechanistic data.

The authors need to explain why the response in the 16Gy portion of the tumor is relevant to outcome. 16Gy is effective alone – it is only when some of the tumor is untreated that treatment fails. Standardly, this will be due to outgrowth of the untreated cancer cells which is generally anticipated and has been well modelled in classic radiobiology textbooks. The addition of 2Gy to the untreated portion is the change that matters. This is really nice data, but the authors seem to be suggesting that the 16Gy treated portion of the tumor is less controlled when a neighboring area is untreated, since it is in this 16Gy region that neutrophils increase. The standard discussions at present, and discussions in the manuscript, talk about low doses modifying the environment of the tumor to permit control. Thus, it would seem that the 2Gy region should show the effects that matter. In the flow cytometry and scRNASeq, the 2Gy region of the 16Gy/2Gy tumors does not seem very different from the 0Gy region of the 16Gy/0Gy tumors. It is perfectly reasonable that this reviewer has missed the authors' point – but please clarify. The discussion would be fine – perhaps where bilateral communication is discussed.

Minor issues

Representative flow cytometry should be shown as a supplement identifying the gating strategy that leads to each assessed population.

The fold change in populations reported in the flow data should be clarified – are these absolute number changes or fold changes in proportions relative to a parent population?

The authors use both flow cytometry and scRNASeq but do not integrate these data well enough. Are all of the changes seen with flow cytometry recapitulated in scRNASeq and vice versa? Fold changes in T cells are shown in flow cytometry but the graphs shown for scRNASeq do not seem to show this effect. We need to know that these datasets are complementary and if not why not.

CXCR2 blockade is only tested with 16Gy/2Gy. The MC38 model appears to have a very large neutrophil population at baseline. Is neutrophil inhibition a generally positive thing with radiation or is it only relevant when neutrophils increase? Similarly, the Ly6G therapy shown in the supplement has been effective in some models and not in others. If the neutrophil depletion helps every treatment group, then the mechanistic link to neutrophil increases by one treatment versus another becomes much less relevant.

Reviewer #4 (Remarks to the Author):

Summary:

Inspired by the interesting findings on the effects of LDRT on tumor microenvironment's immune infiltration, and the trending synergistic treatment of radiotherapy and immunotherapy, Paul et al. in this manuscript proposed a novel approach based on the joint effort of LDRT, HDRT, and ICI. The authors demonstrated the potential of the proposed method using a murine model of colorectal cancer under various conditions. Furthermore, they compared the profiles of immune cells in different tumor microenvironments to further investigate the potential mechanism underlying the observed efficacy. The paper in my opinion is useful as a proof of concept for the proposed treatment and could serve as a basis for future research. However, I have some concerns about the results of the analyses.

Major comments:

- Line 169-170 "the samples from 16/2 irradiated tumors clustered separately from the T116

and TI2 samples". However, from Figure 3C what the authors claimed is not entirely clear. It seems that the clustering is largely determined by the actual dose the tissue received. Control is mixed with the 0Gy group, TI2 is mixed with the 2Gy group, etc. However, within each cluster, samples seem to be mixed with some subtle difference among groups. What's more, as the analysis is conducted using 44 variables with only 40ish samples, I do wonder if the observed difference or the classification result from PLSDA is due to the small sample size.

- While determining a functional role of CD8 T cells in 16/2 irradiations groups, the authors depleted those cells and measured the resulting effects on tumor volumes and survival. However, given the critical role of CD8 T cells, one would maybe expect to observe a similar phenomenon had the authors repeated the same experiment on the TI group. If this is true, the data here might just have reiterated the importance of CD8 T cells regardless of the treatment.

- The conclusions drawn from KM survival curves could benefit from maybe log-rank tests to see if the effects are statistically significant. Specifically, the effect of CD8 T cell depletion and the effect of SB225002.

Minor comments:

- The acronyms: IR (line 48) and DC (line 51) (I guess might stand for dendritic cell?) seem to have appeared without their respective definitions (on line 67 IR was then defined). It might be helpful to spell out what they stand for before using the acronyms.

- Several figures could potentially benefit from using a more contrasting color scheme.

Specifically, Figure 1C (maybe even consider different line types), dots for PI16/* groups and TI16 group

- The legend for Figure 1F appears to be D

- The legend for Figure 2D mentioned "the efficacy experiment shown in E-F". Is "E-F" referring to Figure 1E-F?

- Data and analysis code for generating the results should be deposited in a public repository for transparency and reproducibility. If the data cannot be released to the public before the manuscript is published, data could potentially be uploaded to a repo like GEO and then released afterwards.

We would like to thank the reviewers for considering our work and for the fair suggestions to improve our manuscript. Please find below our point-by-point response (in italic):

Reviewer #1 (Remarks to the Author):

Even though being ignored for decades, it has become obvious that targeted radiotherapy (RT), in addition to the frequently mentioned immune-suppressing effects, has also some immune-active ones. Currently, a paradigm shift takes place that suggests more individual radiation applications regarding dose and volume of irradiation when considering, besides the local effects of RT on tumor cell clonogenicity, also the local and systemic immune modulatory consequences. Most likely RT schemes will have to be adapted in the future and more individualized concepts will be needed.

The paper of Bergeron et al. picks up on one of these possible very promising innovative approaches. They hypothesized that making use of both, killing of tumor cells by high dose RT in combination with immune stimulating effects of lower doses of RT should be beneficial for induction of anti-tumor immune responses, at least with additional immune modulation. Regarding the latter, the approach to test inhibition of the PD-1/PD-L1 axis is well comprehensible, as RT has shown in various settings to increase expression of the immune suppressive checkpoint molecule PD-L1 and partly also PD-L2 on tumor cells. They focus on colorectal cancer as model system (MC38 tumors ectopic in C57BL/6 mice) and focus on irradiation parts of one tumor mass with both, 2Gy and 16Gy RT. This requires a very precise irradiation of parts of the tumor and could be achieved clinically with SBRT.

In their preclinical settings it was achieved to perform irradiations with millimetric precision to deliver different irradiation doses. They found that irradiation of one tumor mass with both low and high dose RT resulted in infiltration of CD8+ T cells, what is generally known, but also of neutrophils, being a new discovery. Neutrophil count increase is known as negative predictor for immune therapies of solid tumors and the new finding of Bergeron and colleagues shed light into possible mechanisms: CXCR2 was identified as a target to reduce neutrophil infiltration. In addition, single-cell RNA-seq revealed that this new irradiation setting reshapes the tumor-infiltrating CD8+ T cells into a more cytotoxic phenotype. At least in their model system a multimodal therapy of partial irradiation with low and high doses in combination with inhibition of PD-1 and CXCR2 was effective in inducing anti-tumor immunity.

To further stress the new findings the authors should consider the following:

Please already define in the abstract why a single dose of 2Gy is a "low dose", since this dose is the standard single dose of "high dose RT" for solid tumors:

In the context of radioimmunotherapy, single irradiation doses ranging from 0.5 to 2 Gy have been described as 'low doses', as they are considerably lower than "ablative" high doses

delivered as single fraction or in hypofractionated regimens. The definitions of low doses radiotherapy (LDRT) or low-dose irradiation (LDI) have been widely used in previous literature to indicate doses ranges of 0.5-2Gy in preclinical settings (Klug et al. 2013; Ochoa de Olza et al., 2020; Herrera et al., 2022; Li et al., 2023). We have now added a sentence in the abstract (lines 29-30) and in the discussion (line 326 and 336-339) clarifying the doses of 0.5-2Gy have been already named LDRT and described as able to reshape the TIME towards a more antitumor one.

We have also better clarified this point in the introduction (see our answer in point 3).

What was the rationale for choosing 16Gy as high dose?

We chose an irradiation dose that was sufficient, in a single administration, to induce complete responses in the model that we first characterized, i.e. the MC38 subcutaneous tumors, according to previous data from our lab. The data reported in Fig. 2A-C confirms that 70% of the mice had complete responses after a T116 irradiation. Moreover, 16 Gy has already been considered as ablative in various cancer radiotherapy-oriented studies from the literature (Antoni et al., 2014; Sahgal et al., 2021; Rostami et al., 2022; Aboussekhra et al., 2023). We clarified the choice of this dose in the discussion section (line 323).

In the introduction LDRT has to be defined more detailed, as the term LDRT is also used for irradiation of chronic inflammatory and degenerative diseases with as single dose per fraction of 0.5Gy. This should be discussed and respective literature be cited.

We agree with the reviewer that the definition of LDRT could be misleading as it is used in different pathological settings, i.e. oncology and inflammatory/neurodegenerative diseases. In the context of solid tumors, LDRT has been used in the range of 0.5-2Gy (see point 1) to favor an antitumor immune response. In contrast, in the frame of inflammatory and neurodegenerative diseases, LDRT at doses of ≤ 0.5 Gy has been used as anti-inflammatory. We have now added a paragraph in the discussion section to address this point (citing relevant literature), stating that further studies will be needed to evaluate what is the optimal combination of doses to achieve an improved antitumor immune response, and in particular in terms of LDRT.

It would have been nice to validate the results in an orthotopic setting. However, it is clear that in a first setting focus has to be set on precise irradiation and that's why the ectopic approach might be more elegant. The data in Figure1 even prove biologically with γ H2AX staining that one part of the tumor was exposed with a high dose and the other part with a low dose, confirming the physical irradiation planning and setting. Nevertheless, this should at least be discussed in more detail.

We share the vision of the reviewer concerning the added value of orthotopic models when compared to subcutaneous ones. As correctly observed by the reviewer, we started with subcutaneous tumor as they represent an easier setting to perform partial irradiations and subsequent analyses with the required precision. In the revised version of the manuscript, we

have now included an orthotopic model of triple negative breast cancer (4T1 cells) in which we validated the usefulness of PI16/IR in combination with aPD1 (supp Fig. 1C-E), and a trend for a better response when CXCR2 inhibitor was used (Supp Fig. 6). We agree with the reviewer that further studies are needed in orthotopic settings, including for rectal cancer, and we are already working on head and neck and lung settings. Such settings will also allow us to perform toxicity studies. We have added a sentence in the discussion (lines 461-468).

Data of Figure 2 indicate that RT with 16Gy to only one part of the tumor in combination with 2Gy and anti-PD1 is as effective as RT with 16Gy to the whole tumor. It would be nice to see as control also data of the setting of irradiation of the whole tumor with 2Gy alone and in combination with anti-PD-1.

We thank the reviewer for their appropriate suggestion, we have now performed total tumor irradiations at 2Gy, alone or in combination with anti-PD1 in the three tumor models now used. As shown in Figure 2 (see panels B, C, and Supp. Figure 1B) the combination of anti-PD-1 to TI2 only slightly tended to delay MC38 tumor growth, with no complete responses observed in this model. TI2 and PD1 blockade alone only induced a slight tumor control in the CT26 model, while TI2 + anti-PD1 induced 33% of complete responses. In both models, the efficacy of any of these three treatments (anti-PD1, TI2, TI2 + anti-PD1) was significantly lower than PI16/2 + anti-PD1. In the orthotopic setting, which is more refractory to RT and PD1 treatments, TI2 was not effective even in combination with PD1, while PI16/2 resulted in a tumor growth delay and a significantly increased survival.

The limited efficacy of LDRT in our experiments could partly be due to the size of the tumors in the beginning of the treatment (~120-150 mm³), which were larger than the standard tumor sizes that are used in the literature for such preclinical studies. Therefore, these data highlight the usefulness of combining HDRT to LDRT within the same tumor volume, in particular for large, bulky tumors. Thanks to the reviewer's comment, we have strengthened our data suggesting that PI may be a viable option whenever possible. We have included these important results in main figure 2 and in Supp Fig. 1 and discussed them in lines 328-331 of the discussion.

The following mechanistic analyses indicate that partial irradiations with 16Gy and 2Gy create a distinct immune micro-environment being mainly characterized by infiltration of CD8+ T cells, NK cells and neutrophils with an immune suppressive phenotype. It was confirmed by RNA-seq analyses and mainly CD8+ T cells are phenotypically re-shaped following partial irradiation. Analyses of the neutrophil cluster revealed that these cells could be distinguished as three distinct groups. These results are the basis to optimize multimodal settings by specifically targeting these immune cells. Depletion of neutrophils improved tumor control and mouse survival when combined to partial irradiations of 2Gy and 16Gy and anti-PD1. Further, using a selective CXCR2 antagonist revealed that it prevented the neutrophil increase. The advantage in comparison to whole irradiation with 16Gy might be less toxicity. However, data about toxicity are missing and should be included or at least being discussed in detail.

The reviewer legitimately points out the crucial matter of toxicity in this setting. One of the main goals of this article was to obtain the proof-of-concept of the usefulness of intratumor IR dose modulation in a simple setting that allows performing precise irradiation under strictly controlled conditions (subcutaneous tumors), as correctly observed by this reviewer (point 3). However, subcutaneous tumors are not an ideal setting to study RT toxicity. Further studies will be conducted in the future using orthotopic models (see our answer to point 4) to assess the efficacy and the toxicity related to PI.

In such settings, it would be conceivable to imagine that toxicity could decrease in PI settings compared to total volume HDRT, similarly to what is observed in the frame of spatially fractionated RT (SFRT) (Mahmoudi et al., 2021). Using PI in the clinics would allow to only target tumor areas that are distant from the potential OARs with HDRT, and to target the rest of the tumor (which is closer to OARs) with LDRT, therefore limiting toxicity to said organs. We discussed these points lines 458-463.

**Are the observed effects general ones or just being observed in the MC38 model?
A second model for confirmation would be very valuable.**

According to the suggestion from all reviewers, we now performed efficacy experiments with subcutaneously implanted CT26 tumors to evaluate the effect of PI16/2 + anti-PD1 on tumor growth in a second colorectal cancer model. Results that are shown in Figure 2D-E indicate that PI16/2 + anti-PD1 has a strong anti-tumor effect on CT26 tumors, with ~90% complete tumor clearances (Supplemental Figure 1B). Moreover, PI16/2 + anti-PD1 is close to achieving a significantly better tumor control than PI16/0 + anti-PD1 (p value = 0.0659). These data suggest that also in this "hot tumor" setting, where PD1 alone and in combination with LDRT already exert an antitumor effect, the use of a combination of HDRT and LDRT provides better tumor control when combined to ICI. Finally, we aimed at confirming our data in an orthotopic setting. We selected the 4T1 triple negative breast cancer model injected in the mice fat pads. Such model, despite being orthotopic, is easy to irradiate in PI conditions. Our data clearly indicate efficacy of the PI16/2+PD1 treatment, being largely more effective than TI2+PD1 treatments. Altogether, these data obtained in three different models support the potential translational value of the novel approach here proposed.

Reviewer #2 (Remarks to the Author):

In this study, Bergeron and colleagues use precise irradiation (IR) techniques to gain novel insights into the modulation of the tumor microenvironment (TME) in response to non-uniform radiation doses. Specifically, subcutaneously-implanted colorectal cancer-derived MC38 cells, upon tumor growth, received different IR regimens. Applying high dose (16Gy) on one tumor half and low dose (2Gy) (PI16/2) on the other revealed distinct TME characteristics compared to uniformly irradiated tumors, affecting cytokines, immune cell abundance and phenotype. Importantly, PI16/2 synergized with anti-PD1 treatment with even some complete responses observed, while each treatment modality (anti-PD1 or PI16/2) alone only modestly diminished tumor growth and augmented mouse survival. The study also explored the molecular

alterations and importance of neutrophils, notably by preventing their recruitment through CXCR2 blockade, which synergises with anti-PD1 upon concomitant PI16/2 IR. Overall, the approaches undertaken and data obtained are fundamentally important; the methodology is sound; the data show promise in enhancing the efficacy of radiation and immunotherapy combinations, particularly in cases where whole-tumor irradiation is impractical due to tumor size or proximity to sensitive organs. The findings, in addition to their fundamental insights, could thus have strong clinical value.

I have a few comments, which I hope will be helpful to the authors. Major comments:

1. Although the work done is original and impressive, a potential weakness is its reliance on a single cell line model grown subcutaneously. It would be nice to be able to confirm at least some findings using another tumor model with similar IR (PI16/2) procedures.

We wish to thank the reviewer for their positive comments on our experimental work, and for their interesting suggestions. According to all reviewers and editor's comments, we have now performed key experiments in additional models (including an orthotopic one), confirming the efficacy of PI16/2 in combination with anti-PD1 (see our answer to Rev 1, last point).

2. It is important to validate neutrophil depletion appropriately, because anti-Ly6G often fails to deplete these cells, depending on the protocol used, mouse age + background, and because many reports from the literature use similar antibodies for depletion and for validation, without considering antigen masking. Here, how was neutrophil depletion validated (Fig. S4A)? The authors should be careful about this point, especially as anti-Ly6G induces neutrophil turnover, so younger cells could be released from the bone marrow and influence the phenotype observed. Some useful information may be found in Nature Com 2020 (PMID: 32488020).

According to the reviewer's suggestions, we have now shown in Supplemental Figure 5 the cytofluorimetric analysis showing the confirmation of tumor neutrophils depletion. As we used anti-Ly6G antibodies for neutrophil depletion, we decided to use Gr-1 as a marker for the identification of neutrophils in this validation experiment, to avoid the epitope masking of Ly6G. The neutrophil depletion through Ly6G was validated using the following gating strategy: cells that were CD45⁺; CD11b⁺; Gr1⁺; Ly6C^{low-int} were considered as neutrophils (see gating strategy in Supp Fig. 5A). We discussed this point line 281, citing the suggested literature.

Concerning the neutrophil turnover, we agree with the reviewer that this is an important point to consider when performing neutrophil depletion experiments. Nevertheless, in the current study, our goal was to deplete neutrophils in a short time window, corresponding to the observed increased neutrophilic infiltration two days after irradiation. As we performed only one anti-Ly6G injection (on the day of irradiation), we agree that we cannot conclude about the effect of long-term depletion.

3. Linked to Figure 5: CD8 T cell phenotypes are only inferred from transcriptional

signatures. It would be interesting to test or validate some changes at the protein level, to increase confidence about their effector functions. Specifically, do CD8 cells upon PI16/2 IR express different levels of cytoplasmic IFN γ or TNF upon in vitro reactivation (PMA+ionomycin)?

According to the reviewer's suggestion, flow cytometry experiments following in vitro activation with PMA + Ionomycin have been performed with tumors treated with PI16/2 and/or anti-PD1 (as requested by reviewer 3). The results of flow cytometry evaluation of the number of CD8⁺ T cells expressing IFN γ or Granzyme B (which is crucial for their effector functions) can be found in Supplemental Figure 3E. We observed that CD8⁺ T cells expressing IFN γ or Granzyme B tended to increase in PI16/2 tumors compared to the Ctrl condition, with this difference further increasing and becoming significant when anti-PD1 is combined to PI16/2 (in the PI16/2 2Gy portion of the tumor). These results are in line with an increased CD8 activation following PI16/2 irradiation and anti-PD1, supporting their role in the antitumor response observed following this combined treatment.

Minor comments:

1. Validate CD8 T cell depletion efficacy.

The CD8 T cell depletion protocol used in this study has been proven to be effective in previous studies by our group (Mondini et al, 2015 and 2019, Hamon et al., 2022), and by others (Jin et al., 2022). Here below the analysis of depletion efficacy from another study conducted in parallel with the present one by our group, using an identical protocol, validating an effective depletion of CD8 T cells in MC38 tumors up to three days after antibody injection, in line with the clear effect of CD8 T cell-depletion observed in Figure 6A-C.

[figure redacted]

2. The authors mention that the expression of 44 cytokines was tested. It seems some information is lacking as of the technique used and the identity of the other cytokines. Were they undetected, or not significantly changed?

The level of expression of 44 cytokines was assessed by cytokine profiling based on the Illumina technology by the Eve Technology company, following the protocol described in the "cyto/chemokine array" paragraph of the Methods section. The list of cytokines tested is present on the company website, but for clarity reason, we now included it in the methods section (line 618) The levels of 35 of these 44 cytokines (the others were excluded as their

expression level was outside the dynamic range of the assay) were used to generate the Partial Least-Squares Discriminant Analysis (PLS-DA) that can be found in Figure 3C. Among these cytokines, the detailed quantifications of CCL4, CCL5, CXCL1, CXCL2 were shown, as they can be related to neutrophils function/phenotype. Most of the other cytokines were not significantly changed. We have now included all the data concerning the relative levels of the cytokines in the "raw data file" uploaded with this resubmission.

3. Linked to Figure 3D, it would be interesting to identify which cell types produce the differentially expressed cytokines. Are these tumor cells, or immune cells, perhaps directly neutrophils? Also, was Cxcl5 expressed and induced by PI? Finally, kinetics of cytokine measurements and neutrophil phenotypic analyses would be interesting, but perhaps for future follow-up studies.

We agree with the reviewer that these observations could provide informative insights into the modulation of TIME. Nevertheless, we could not argue what cell types are the sources of the differentially regulated cytokines from our bulk cytokine profiling. Some clues can be obtained by our scRNAseq analysis, limited to the immune cell populations. Neutrophils express high levels of CCL4 and CXCL2, while transcriptomic levels of expression of CXCL1 and CXCL5 were too low to be considered. CCL5 was highly expressed in DCs (see below).

[figure redacted]

As these analyses are limited to a few cytokines and cell types, we hope that this reviewer agrees that they are too preliminary to be included in this manuscript, and left for future studies, as for the analysis of their kinetics, as suggested. We added a comment in the discussion, line 364.

4. Figure 3B: reviewer wonders if the scale “fold increase” should not be called instead “fold change”, as decreases are also observed in comparison to control. Also, is it correct to start at “0”, while a fold increase of 0.5 should equal to -2, and as, with this scale, a decrease of 2-fold appears smaller, visually, to an increase of 2-fold.

We agree with the reviewer that the definition of Y axis was incorrect. We have now changed it to “Relative level” and explained in the figure legend that data have been normalized to the control level from each panel.

Since there is no significant decrease observed, we preferred to leave the graph as it is (with the updated Y axis definition), as it is more commonly used.

5. Figures 2CD+6AB+S4BC: statistics are missing.

Following the reviewer’s comment, we have now shown the results of the statistical analyses in the indicated figures.

6. Legend of Figure 1: Replace ‘D’ by ‘F’.

We have corrected the panel lettering.

7. Line 231, reviewer is unsure of the meaning “Corresponding to different degrees of plasticity”, when referring to the three neutrophil clusters.

We agree with the reviewer that such definition was misleading, we have now changed it to “Corresponding to different neutrophil phenotypes” (line 259).

Reviewer #3 (Remarks to the Author):

Bergeron et al NCOMMS-23-48193 Non-homogenous intratumor ionizing radiation doses synergize with PD1 and CXCR2 blockade

Summary

The manuscript describes the impact of reducing the tumor area that is irradiated with a higher dose and adding low dose radiation to the remaining tumor. The results nicely show that while a total high dose of 16Gy is effective, only irradiating half of the tumor with this dose results in a loss of tumor control. Adding anti-PD1 to this partial treatment is poorly effective. Irradiating with a combination of 16Gy and 2Gy results in improved control over 16Gy combined with no dose when combined with anti-PD1 therapy.

The authors see a clinical rationale for reduced dose, where there are limited opportunities for high doses to all of the tumor due to risk of toxicities. However, a split dose is less effective than a total high dose in this tumor, so while they show that giving some irradiation to the

remainder is better than nothing, it does not provide evidence that split doses are optimal. In fact, the 16/2Gy treatment plus anti-PD1 remains less effective than 16Gy to all of the tumor without anti-PD1, and in this MC38 model doses in the 16Gy range plus anti-PD1 are generally curative. Nevertheless, the data does clearly demonstrate a value to adding some dose over no dose to the parts of the tumor that do not receive 16Gy.

Analysis of the tumor with flow cytometry and with scRNASeq show some T cell and neutrophil correlates with the 16Gy/2Gy response, but these are not very consistent. CD8 T cells are shown to be necessary for tumor control by RT plus anti-PD1, which is expected. Neutrophil increases are seen in the 16Gy/2Gy treatment and not seen in the 16Gy/0Gy therapy, and postulated as a limitation in tumor control. However, neutrophils also increase in 16Gy total dose, which is effective. In addition, it is not clear why anti-PD1 is not included in these studies. 16Gy/2Gy is very similar in tumor control to 16Gy/0Gy in the absence of anti-PD1, so the neutrophil change cannot be postulated as a reason for the different outcome – there is no different outcome in the samples where neutrophils change.

The manuscript is well written, the figures are clear and the conclusions are generally valid. However, the limitations discussed above and below reduce the overall impact of the work.

Major issues.

This is a single tumor model. It is not reasonable to infer too much from these results in MC28 colorectal carcinoma unless they are validated in additional tumor models.

We agree with this comment of the reviewer, and according to their suggestions f, we have now performed additional experiments in a second colorectal tumor model, and in the orthotopic model of breast cancer 4T1. The results obtained strongly support the possibility to use PI in combination with anti-PD1 (see our answer to reviewer 1, last point).

The CXCR2 inhibition does not improve tumor responses when combined with 16Gy/2Gy treatment; it only improves responses when also combined with anti-PD1. The changes in neutrophil differentiation associated with 16Gy/2Gy treatment may have nothing to do with the mechanism, as we do not know what happens when anti-PD1 is also present, where the different response is seen. Since 16Gy/2Gy treatment changes neutrophil maturation and yet CXCR2 inhibition does not improve tumor responses unless anti-PD1 is present, the proposed mechanism is not proven. The authors need to model the response – show what is different when anti-PD1 is combined with 16Gy/2Gy versus 16Gy/0Gy. Clearly, CXCR2 inhibition has an effect here, but is this actually correlated with the maturity effect observed without anti-PD1?

Our hypothesis was that the use of PI16/2 might provide a tumor immune environment more

prone to the reactivation by anti-PD1, and our data seemed in line with this observation. We agree with the reviewer that it is of particular interest to evaluate the neutrophil levels/phenotype after PD1 treatment. We thus performed flow cytometry analyses to assess the infiltration of CXCR2⁺ neutrophils in tumors treated with PI16/2 + anti-PD1 (Supplemental Figure 5F). We confirmed an increased level in tumors treated with PI16/2, even if it did not reach statistical significance likely due to the small number of samples in this experiment. Of great interest, the neutrophils number was even higher in tumors treated with PI16/2+aPD1, reaching a statistical significance in the portion of the tumor irradiated at 16Gy (left panel). Such neutrophils were mostly CXCR2 positive. Accordingly, the number of CXCR2⁺ neutrophils was significantly increased in the 16Gy portion of PI16/2+aPD1 tumors (right panel). In addition, we observed and increased effector function of CD8 T cells in tumors treated with PI16/2+aPD1 (see our response to reviewer 2, point 3, and Supp Fig. 3E)

The observation that CD8 T cell depletion eliminates tumor control by 16Gy/2Gy combined with anti-PD1 is not sufficient to validate the authors' conclusions on CD8 T cell differentiation in the tumor. That anti-PD1 functions via CD8 T cells is evident and very well demonstrated in this and other models. If the authors aim to link the changes in T cell differentiation in the tumor to treatment responses, then more targeted interventions are needed to determine whether these slightly different populations are important. Otherwise these observations that T cell differentiation is slightly different remain correlations and not mechanistic data.

Our data clearly show that T cells display a transcriptomic phenotype which is greatly different in 16/2 tumors when compared to the other treatments. Moreover, our novel data obtained by flow cytometry analyses after PD1 treatment suggest an increased effector function of these cells in PI16/2+aPD1 tumors. Finally, our data using aCD8 depleting antibodies reiterate the importance of CD8 T cells in PI16/2 IR settings (see our response to Reviewer 4, point 2).

Nevertheless, we completely agree with the reviewer that our available data do not allow us to assign a role to specific CD8 subsets, as they are identified by scRNAseq. Defining and performing targeted strategies to deplete/remodulate specific CD8 subset would require efforts that are beyond the main scope of this paper (ie. proposing the proof of concept about the potential interest of a novel IR modality), and not achievable in the time frame of a revision. We have thus modified the results and the discussion sections to avoid any overstatement, and clearly stating that our observations concerning the CD8 T cell phenotype are correlative.

The authors need to explain why the response in the 16Gy portion of the tumor is relevant to outcome. 16Gy is effective alone – it is only when some of the tumor is untreated that treatment fails. Standardly, this will be due to outgrowth of the untreated cancer cells which is generally anticipated and has been well modelled in classic radiobiology textbooks. The addition of 2Gy to the untreated portion is the change that matters. This is really nice data, but the authors seem to be suggesting that the 16Gy treated portion of the tumor is less controlled when a neighboring area is untreated, since it is in this 16Gy region that neutrophils

increase. The standard discussions at present, and discussions in the manuscript, talk about low doses modifying the environment of the tumor to permit control. Thus, it would seem that the 2Gy region should show the effects that matter. In the flow cytometry and scRNASeq, the 2Gy region of the 16Gy/2Gy tumors does not seem very different from the 0Gy region of the 16Gy/0Gy tumors. It is perfectly reasonable that this reviewer has missed the authors' point – but please clarify. The discussion would be fine – perhaps where bilateral communication is discussed.

We thank the reviewer for pointing out that our comments concerning the mechanism underlying the efficacy of PI16/2 were not clear. We agree with the reviewer that the 16Gy portion is likely well controlled no matter what the treatment in the neighboring volume is, and it was not our intention to suggest that it is "less controlled when a neighboring area is untreated". We do agree that the changes occurring after LDRT are those that matter. We changed our comments in the discussion section, highlighting that the changes observed in the 2Gy region of PI16/2 vs the untreated regions in PI16/0 (see Figure 3B, Figure 4C, Figure 5C-E) can contribute to the different outcome observed in these two treatment groups. Moreover, as there is likely interplay between the different regions of the tumors (as correctly observed by this reviewer), the differences in immune infiltration in the 16Gy at day 2 can probably play a role as these cells can traffic to the other region, especially during the tumor regression of the HDRT volumes. These points are now discussed in lines 361-363 and 408-412.

Minor issues

Representative flow cytometry should be shown as a supplement identifying the gating strategy that leads to each assessed population.

We have now shown the gating strategy for lymphoid cells in Supp Fig. 2A, while for myeloid cells it is published in Gerbé de Thoré et al 2023.

The fold change in populations reported in the flow data should be clarified – are these absolute number changes or fold changes in proportions relative to a parent population?

According to the comments from review 2, we have now changed the axes to "Relative levels" and explained that such levels were calculated on absolute number changes in the figure legends.

The authors use both flow cytometry and scRNASeq but do not integrate these data well enough. Are the all of the changes seen with flow cytometry recapitulated in scRNASeq and vice versa? Fold changes in T cells are shown in flow cytometry but

the graphs shown for scRNASeq do not seem to show this effect. We need to know that these datasets are complementary and if not why not.

In our flow cytometry analyses, the levels of the different immune populations come from absolute cell numbers detected in the tumors. On the other hand, scRNAseq data represents proportions of CD45+ cells. Thus, one cannot perform direct comparisons, as the changes of numerically more represented populations (ie. neutrophils and monocytes/macrophages) outweigh the less abundant ones (eg. lymphocytes). Accordingly, neutrophils data from the two techniques are mostly in agreement (compare Fig 3B and 4C for neutrophils), while the proportion of lymphocytes is reduced in PI16/2 samples when analyzed by scRNAseq, likely due to the large increase of neutrophil population in this treatment group (compare control and PI16/2 samples in fig 4C). These data highlight the importance of performing absolute cell counts whenever possible, especially when analyzing less abundant populations. We have included a comment in the discussion section (lines 345-347).

CXCR2 blockade is only tested with 16Gy/2Gy. The MC38 model appears to have a very large neutrophil population at baseline. Is neutrophil inhibition a generally positive thing with radiation or is it only relevant when neutrophils increase? Similarly, the Ly6G therapy shown in the supplement has been effective in some models and not in others. If the neutrophil depletion helps every treatment group, then the mechanistic link to neutrophil increases by one treatment versus another becomes much less relevant.

We agree with the reviewer that MC38 tumors appear to have large basal neutrophil infiltration, which is increased, even if at different extents, in all HDRT treated tumor tissues. It would be thus conceivable to speculate that CXCR2 inhibition could have an effect in all irradiated samples. In line with this view, previous literature findings indicated that CXCR2 inhibitors can increase the efficacy of RT (see eg. PMID: 34124076). Further studies, beyond the scope of the current article, would be needed to validate such observations in our models. We added a comment in the discussion section.

Reviewer #4 (Remarks to the Author):

Summary:

Inspired by the interesting findings on the effects of LDRT on tumor microenvironment's immune infiltration, and the trending synergistic treatment of radiotherapy and immunotherapy, Paul et al. in this manuscript proposed a novel approach based on the joint effort of LDRT, HDRT, and ICI. The authors demonstrated the potential of the proposed method using a murine model of colorectal cancer under various conditions. Furthermore, they compared the profiles of immune cells in different tumor microenvironments to further investigate the potential mechanism underlying the observed efficacy. The paper in my opinion is useful as a proof of concept for the proposed treatment and could serve as a basis for future research. However, I have some concerns about the results of the analyses.

Major comments:

- Line 169-170 "the samples from 16/2 irradiated tumors clustered separately from the TI16 and TI2 samples". However, from Figure 3C what the authors claimed is not entirely clear. It seems that the clustering is largely determined by the actual dose the tissue received. Control is mixed with the 0Gy group, TI2 is mixed with the 2Gy group, etc. However, within each cluster, samples seem to be mixed with some subtle difference among groups. What's more, as the analysis is conducted using 44 variables with only 40ish samples, I do wonder if the observed difference or the classification result from PLS-DA is due to the small sample size.

We agree that the initial formulation was a little too simplistic. It was replaced by:

"The samples from 16/2 irradiated tumors were projected in the same sPLS-DA regions than the TI16 and TI2 samples respectively but in specific sub-regions (right for PI16/2-16Gy and top for PI16/2-2Gy)."

The reviewer is right about the structure of the dataset: the number of samples and the predictors have a similar order of magnitude and caution is needed when performing statistical analysis since machine learning algorithms tend to overfit these highly dimensional datasets which impact their generalizability on new data.

In our dataset, many cytokines can provide redundant information about the biological process of interest and can therefore be summarized by the smallest number of new covariates obtained after a dimension reduction approach. Thus, a sparse partial least square discriminant analysis (sPLS-DA) is a suitable approach for both dealing with multicollinearity and performing an efficient cytokines selection.

In this study, a sparse dimension reduction to two dimensions was conducted. Thanks to the variable selection ingredient, two new covariates were obtained as a linear combination of only four cytokines (IL10, LIX (CXCL5), MIP.1b (CCL3) and Rantes (CCL5)) and defined a new planar space where the model compute two coordinates (variate 1 and variate 2) for each sample (animal) for visualization (Figure 3C).

In all the analyses, leave-one-out cross validation was used to determine the model parameters including number of components and number of features per component. Cross-validation is a model validation technique used to assess whether the results of an analysis can be generalized to another data set. It consists in dividing the data set into folds, training the model on all folds except one and evaluating the prediction performance on the left-out subset. This process is iterated until each subset is left out once; in the case of Leave-One-Out CV used in this study, each fold consists of one sample which is left out once.

Therefore, we believe that the obtained classification results can be considered as robust and unbiased.

- While determining a functional role of CD8 T cells in 16/2 irradiations groups, the authors depleted those cells and measured the resulting effects on tumor volumes

and survival. However, given the critical role of CD8 T cells, one would maybe expect to observe a similar phenomenon had the authors repeated the same experiment on the TI group. If this is true, the data here might just have reiterated the importance of CD8 T cells regardless of the treatment.

We agree with the reviewer's comment and following his suggestion we have performed another experiment using anti-CD8 antibodies in the TI group. Our data show that the efficacy of the 16/2+/- aPD1 treatment is completely abrogated when CD8 T cells are depleted. Conversely, CD8 T cells are partly dispensable in TI16 irradiation settings, as we observed a delayed tumor growth and a significantly increased survival in the TI16+aCD8 group, when compared to control mice (figure 6A-C). These observations reiterate the pivotal role of CD8 T cells for the response of the PI16/2 treatments. Differently, in TI16 settings the cytotoxic effect of this ablative dose seems to play a major role for the antitumor response, even if the contribution of CD8 activity is still observed, especially to achieve complete response (see Fig. 6B).

We have updated the results accordingly and added this comment in the discussion section (lines 352-356).

- The conclusions drawn from KM survival curves could benefit from maybe log-rank tests to see if the effects are statistically significant. Specifically, the effect of CD8 T cell depletion and the effect of SB225002.

Statistical analyses using the Log-rank test have now been included in the figures. They strongly support the antitumor activity of HDRT+LDRT in combination with PD1, not only when compared to control group, but also vs immunotherapy alone and vs total LDRT+PD1, in all the three tumor models used (see Fig. 2 and Supp Fig. 1). The analysis also allowed us to validate the different effect of CD8 depletion in PI16/2+aPD1 and TI16 settings (see Fig. 6). Concerning the effect of SB225002, the analyses did not allow to identify statistical differences for survival in PI16/2+aPD1+SB225002 versus PI16/2+aPD1 groups, even if a trend was observed and the difference was remarkably consistent along the biological replicates performed. This may be explained, at least in part, by the already high survival rate of PI16/2+aPD1 mice bearing MC38 tumors. We have now added some comments in the discussion section indicating that further studies are needed to validate the trends observed (lines 432-435).

Minor comments:

- The acronyms: IR (line 48) and DC (line 51) (I guess might stand for dendritic cell?) seem to have appeared without their respective definitions (on line 67 IR was then defined). It might be helpful to spell out what they stand for before using the acronyms.

We thank the reviewer for this comment. We have defined the acronyms accordingly.

- Several figures could potentially benefit from using a more contrasting color scheme. Specifically, Figure 1C (maybe even consider different line types), dots for PI16/* groups and TI16 group

Curves from Figure 1C were modified using more contrasting color schemes, as suggested by the reviewer.

- The legend for Figure 1F appears to be D

We have corrected the figure legend.

- The legend for Figure 2D mentioned "the efficacy experiment shown in E-F". Is "E-F" referring to Figure 1E-F?

We apologize for the mistake from our side, "E-F" was actually referring to "A-B" on this same Figure 2 (the individual tumor volume curves in Figure 2A and the mean tumor volume in Figure 2B). We have now corrected the figure legend.

- Data and analysis code for generating the results should be deposited in a public repository for transparency and reproducibility. If the data cannot be released to the public before the manuscript is published, data could potentially be uploaded to a repo like GEO and then released afterwards.

As per reviewer and publisher request, data and analysis codes have been deposited in the GEO repository GSE262699.

REVIEWERS' COMMENTS

Reviewer #1 (Remarks to the Author):

The authors have very carefully implemented the reviewers' suggestions and validated their results with additional mouse models, including an orthotopic breast cancer model. In my view, there are no further points of criticism.

Reviewer #2 (Remarks to the Author):

The authors have now responded to my comments satisfactorily.

Reviewer: Etienne Meylan

Reviewer #3 (Remarks to the Author):

The authors have been highly responsive to review and addressed each of this reviewer's points. The additional models, controls, and analyses all improve the author's conclusions and add to the manuscript. This reviewer appreciates the authors' efforts.

No further issues.

Reviewer #4 (Remarks to the Author):

In this version, the authors carefully addressed my previous questions. The added data and model have further enhanced the quality of the manuscript. I do not have any other comments.